# Storm frequency, magnitude, and cumulative storm beach impact along the U.S. east coast

Rachele Dominguez[1], Michael S. Fenster[2], John W. McManus[3]

[1]Department of Physics, Engineering, and Astrophysics, Randolph-Macon College, Ashland, VA 23005, USA
[2]Geology/Environmental Studies Program, Randolph-Macon College, Ashland, VA 23005, USA
[3]Department of Computer Science, Randolph-Macon College, Ashland, VA 23005, USA

*Correspondence to*: Rachele Dominguez (racheledominguez@rmc.edu)

**Abstract.** This study extracted historical water level data from 12 National Oceanographic and Atmospheric Administration tide gauge stations, spanning the period from the early 20th century to 2022 from central Maine to southern Florida, in order to determine if temporal and spatial trends existed in frequency and magnitude of storms along the U.S. Atlantic Ocean coast. We used the Storm Erosion Potential Index (SEPI) to identify and quantify storms. We then use the timing and magnitude of those storms to determine the cumulative effect of storm clustering and large magnitude storms on sandy beaches using the cumulative storm impact index (CSII) empirical model. The results from this study showed (1) no appreciable increase in storm frequency at any of the stations (except for sheltered stations susceptible to storm tide augmentation); (2) statistically significant, but modest increases in storm magnitudes over time for eight of the 12 tidal stations; (3) regional differences in storm magnitudes (SEPI) and cumulative storm impacts (CSII) characteristic of more frequent extratropical storms (temporal clustering) in the north and less frequent tropical storms in the south; and (4) a four to 10 year recovery period for regional beach recovery.

## 1 Introduction

Questions about storm frequency and strength that impact our world's coasts loom in light of climate-induced storm impacts on physical, ecological and human socio-economic coastal systems. Answers to these questions have come from synoptic studies that model oceanographic and meteorologic dynamics and/or studies that seek to quantify the oceanic/coastal responses to meteorologically induced disturbances along coasts using various metrics (e.g., Zhang et al., 2001; Komar and Allan, 2008; Harley, 2017). Regional and global climate models analyze storms using equations that describe the physical laws that govern the interactions among the atmosphere, oceans, land, and ice to simulate future projected outcomes (e.g., Leckebusch and Ulbrich, 2004; Leckebusch et al., 2006; Martin et al., 2006; Muis et al, 2023). Empirical studies use measurable historical proxies or metrics to provide insight into the formation, evolution, frequency, intensity, and duration of storms (e.g., Bengtsson et al., 2007; Knutson et al., 2007, 2008; Gualdi et al., 2008; Zhao et al., 2009; Bender et al., 2010). Thus, empirical studies tend to focus on quantifying storminess using parameters such as wave height, wave power (or energy), water level (storm

tide, storm surge), and/or storm tracks (e.g., Bigelow and Moore, 1897; Hosler and Gamage, 1956; Klein, 1951, 1958, 1965; Hayden, 1975, 1981, 1999; Lozano and Swail, 2010; Meyer and Gaslikova, 2024).

Because of complexities in modelling the thermodynamic responses of storms to global warming, and model domain and resolution limitations (Vecchi et al., 2008; Shaw et al., 2016; Knutson et al., 2020; Sobel et al., 2021) extreme storm simulations

offer inconsistent conclusions about storm *frequency* projections but tend to agree more on storm *magnitude* forecasts. Several studies and reviews, including the review provided by the Intergovernmental Panel on Climate Change's Sixth Assessment Report, indicate that the total global frequency of tropical storms will decrease or remain constant (Tory et al., 2013; Knutson et al., 2020; Seneviratne et al., 2021). In fact, Knutson et al. (2020) indicated that 22 of 27 modelling studies at the time of their publication projected decreasing global tropical cyclone frequency. Others suggest that no clear trends or explainable

variation exist in tropical storm frequency (Sobel et al., 2021). However, some models predict an increase in storm frequency (Camargo, 2013; Emanuel, 2013; Murakami et al., 2014; Wehner et al., 2015; Bhatia et al., 2018; Sobel et al., 2021), especially in the more intense category 4 and 5 tropical cyclones (Bender et al., 2010; Emanuel, 2013; Knutson et al., 2020).

More consensus exists in the modelling results that project an increase in mean tropical cyclone peak intensity and significant

global increases in the proportion of major tropical cyclone intensities driven by warmer sea surface temperatures, especially as shown by finer resolution models (Knutson et al., 2010, 2015, 2019; 2020; Murakami et al., 2015; Walsh et al, 2015, 2016; Wehner et al., 2015, Zarzycki and Ullrich, 2017; Roberts et al., 2020; Seneviratne et al., 2021). Based on a synopsis of 11 studies, Knutson et al. (2020) showed that projected changes in tropical cyclone maximum intensities will increase in the North Atlantic basin by about 3% compared to a median increase for all global basins of approximately 5% and that globally, category

4 and 5 tropical cyclones will increase by about 13%. With respect to extratropical storms, Seneviratne et al. (2021) suggest that current climate models lack the resolution to provide a high degree of projection confidence and therefore, underestimate extratropical storm intensity. However, Colle et al. (2013) found the best (highest resolution) Coupled Model Intercomparison Project (CMIP5) models projected a 5-10% increase in cyclones and 5%-40% increase in deeper cyclones (<980 hPa) by the late twenty-first century inland along the U.S. east coast, but a 15%–20% decrease in cyclones and little increase in strong

cyclones over the east coast water domain. Zappa et al. (2013), Chang (2014, 2018), and Seiler and Zwiers (2016) found similar decreases in extratropical cyclone frequency in various locations within the Pacific and Atlantic basins.

Models have projected additional climate-induced changes to storms will likely include an increase in tropical and extratropical storm rain rates and wind speeds, poleward shifting of storm tracks in both hemispheres, a slowing of cyclone translation speed

and an increase in storm surges, all of which will increase coastal vulnerability (e.g., Kossin, 2018; Yamaguchi et al., 2020; Seneviratne et al., 2021; Muis et al., 2023). Global projections of storm surges based on a high resolution (2.5 km grids)

coupled tide/surge model and a climate hydrodynamic model suggest that median storm surges may increase up to 20% in the future epoch (2021-2050) compared to the historical epoch (1951-1980) although nonuniformly in space (Muis et al., 2023).

Empirical studies use metrics such as best-track position, geographic distributions of cyclones (e.g., mean latitude where cyclones reach their peak intensity), numbers (counts), some measure of magnitude or intensity, translation speed, length in days (track duration), accumulated cyclone energy, power dissipation index, rainfall rates, and rainfall area to assess frequency and intensity of tropical and extratropical cyclones. The heterogeneous nature of historical instrumental data ("best-track" data) and temporal data limitations have made detecting long-term (decadal to centennial) trends in tropical cyclone frequency
or intensity difficult (Schreck et al., 2014; Seneviratne et al., 2021). This finding does not preclude the possibility of the existence of historical trends but speaks more to data reliability. In fact, later studies which have sought to homogenize best-track tropical cyclone data and utilize more recent satellite data are consistent with model projections, i.e., a decreasing number of tropical storms and tropical storm days and increasing intensities with stronger hurricanes (categories 3-5) increasing at a greater rate and a poleward migration of locations where tropical storms reach their peak intensities (Emanuel, 2005; Webster
et al., 2005; Kang and Elsner, 2012; Kishtawal et al., 2012; Kossin et al., 2013; Holland and Bruyère, 2014; Mie and Xie, 2016; Zhao et al., 2018; Tauvale and Tsuboki, 2019; Kossin et al., 2020; Seneviratne et al., 2021). While five of the six global ocean basins displayed increasing tropical cyclone intensity between 1979-2017, the North Atlantic basin had the greatest changes with exceedance probabilities of a major hurricanes increasing by 49% per decade at the 99% confidence level (Kossin et al., 2020). A review of previous studies using proxy data and climate model data to assess storminess in the North Atlantic
showed that storm activity trend analyses strongly depend on the time period used in the analysis and that storm numbers have increased for the most recent decades but show interdecadal variation over centennial time scales (Feser et al., 2014).

Empirical, statistical (probabilistic) and modelling studies designed to couple storm impacts (drivers/forcings) to beach responses have advanced our understanding of coastal system dynamics and hazards although inferences can differ depending
on the specific storm metric(s) used and the coastal impact spatial scale examined. In some cases, these coupled dynamical studies have led to the development of hazard or storm erosion predictive indices (e.g., Thieler and Hammar-Klose, 1999; Stockdon et al., 2007; Mclaughlin and Cooper, 2010; Torresan et al., 2012; Birchler et al., 2015; Leaman et al., 2021; Ahmed et al., 2022 among many). Hamid et al. (2019) provide a review of these indices in three categories: (1) socioeconomic, (2) coastal characteristics, and (3) coastal forcing variables. While some studies have evaluated the direct effects of storms on
coasts using observable and measurable geological (i.e., geomorphological and sedimentological) and/or ecological changes (e.g., Burvingt et al., 2017; Harley et al., 2022), other studies have produced methods to quantify beach erosion *potential* which indirectly evaluates the effects of storms on coastal systems (e.g., Zhang et al., 2001; Miller and Dean, 2006; Miller and Livermont, 2008). With respect to potential erosion, in general, larger magnitude storms or storms clustered in time have a larger beach erosion potential (Karunarathna et al., 2014; Fenster and Dominguez, 2022). Finally, it should be noted that storm

impact studies have investigated storm-driven process-response relationships over a variety of 1D, 2D, and 3D spatial scales including the shoreline, beachface, dune, island, shoreface, littoral zone, and larger reaches, as well as combinations of these scale features (e.g., Morgan and Stone, 1985; Sallenger, 2000; Zhang et al., 2001; Stockdon et al., 2007; Birchler et al., 2015).

      At the spatial scale of the U.S. east coast and centennial temporal scale, natural and potential anthropogenic forcings (e.g., sea-
level rise and storms) threaten increasing populations and coastal development and ecosystems, especially given the geographic position of the U.S. coastline relative to extratropical and tropical storm tracks (e.g., Davis and Dolan, 1994; Friedman et al., 2002; Dinan 2007; Little et al., 2015; Doran et al., 2021). While much is known about the rates, spatial distribution, and acceleration of sea-level rise along the U.S. east coast during the twentieth- and twenty first-centuries (e.g., Sallenger et al., 2012; Ezer, 2013; Ezer et al., 2013; Yin and Goddard, 2013; Harvey et al., 2021; Chi et al., 2023; Yin, 2023) and changes to
the wave climate over decadal time scales (e.g., Davis et al., 1993; Bromirski and Kossin, 2008; and Komar and Allan, 2007), less is known about changes to the storminess (frequency and changes in strength) over longer coastal reaches and time scales – especially using empirical data. Zhang et al. (2000) investigated water level data from 10 tide gauges from Florida to Maine and found no discernible long-term trend in the number and intensity of moderate and severe coastal storms during the twentieth century.


      This study updates Zhang et al. (2000, 2001) Storm Erosion Potential Index (SEPI) assessment of storminess along the U.S. east coast and uses a newly developed cumulative storm impact index (CSII) to account for the timing (clustering) and strength of previous storms, on potential beach erosion along the U.S. east coast (Fenster and Dominguez, 2022). Like Zhang et al. (2000, 2001), we use water level data (storm tide and storm surge) to identify storms (rationale provided in 2.1 Storm
Identification). In particular, we assess the frequency and magnitude of storms along the eastern U.S. coast using historical water level data from 12 tidal gauge stations located from Portland, Maine to Key West, Florida and compare these results to known storm climatology of tropical and extratropical cyclones (e.g., Mather et al., 1964; Zishka and Smith, 1980; Davis et al., 1993; Hirsch et al., 2001; Kossin et al., 2017; Knutson et al., 2020). This study seeks to answer the following questions:

      1. What is the annual storm **frequency** at these locations, how does storm frequency compare geographically, and has
annual storm frequency changed significantly over the time period of record?
      2. What is the total annual **magnitude** of storms at these locations, how does this quantity compare geographically, and has the annual storm magnitude changed significantly over the time period of record?
      3. What is the **cumulative effect** of individual storm magnitudes and frequency (clustering in time) on sandy beaches near these locations?

For this study, we assume, as did Zhang et al. (2001), that the process data obtained from a tide gauge station allows us to make regional-scale (i.e., kilometers) inferences about beach erosion potential along the U.S. east coast. The database and the index used to quantify beach erosion potential (SEPI) used for this study avoid problems associated with heterogenous

historical instrumentation data, relying on hindcast data, and data that only permit frequency analyses and affords analyses over a wide geographic area and long time periods (centennial scale) . In addition, this study addresses questions and concerns about the risk of storm surges increasing in concert with sea-level rise (Ghanavati et al., 2023) – especially along the U.S. eastern seaboard which is especially vulnerable to extratropical storms given the orientation of its beaches relative to extratropical cyclone paths (Davis et al., 1993). A primary advantage of using this method is that sea-level change (i.e., rise) is removed to isolate the impact of storms on beach erosion potential and therefore, a rise in sea-level will exacerbate identified beach erosion potential stemming from storm tides and storm surges.

## 2. Methods

Below, we present our criteria for identifying storms and quantifying storm magnitude using a Storm Erosion Potential Index (SEPI) and a Cumulative Storm Impact Index (CSII) (Zhang et al., 2000, 2001; Fenster and Dominguez, 2022). We then describe the data retrieval process used to acquire water level data from the National Oceanic and Atmospheric Administration (NOAA) Tides and Currents Database (NOAA Tides and Currents) and finally, the methods used to analyze these data.

### 2.1 Storm identification

Following Zhang (1998) and Zhang et al. (2000, 2001), we adopt a Storm Erosion Potential Index (SEPI), defined below, to identify storms and quantify storm magnitude using water level data from tide gauges. These studies have advocated for the use of storm surge (i.e., non-tidal residuals) as a proxy for storm waves, highlighting storm tide and storm duration as the primary factors contributing to beach erosion (Dean, 1991; Zhang, 1998; Zhang et al., 2000; 2001).

Recent studies have shown that wave runup (swash and setup processes) can contribute to extreme water levels and can induce spatially varying erosion impacts along coastlines due to varying continental shelf widths (Stockdon et al., 2007, 2023; Parker et al., 2023). However, Cohn et al. (2018) used new field datasets and a numerical model to show that anomalously high still water levels (caused by storm surge or spring tides) have a greater potential to produce dune erosion than the largest wave energy. Additionally, the effect of storm surge is purported to be larger (and the wave-driven component smaller) on the U.S. east coast than the west coast because the narrower continental shelves on the west coast limit storm surge (and enhance wave energy) more than the wider east coast shelves (Cohn et al., 2018). Serafin et al. (2017) found that slight increases in wave runup and a doubling of storm surge contribute to increases in extreme total water level events and make the case that the storm surge (high-frequency residuals) can have a 10-fold greater effect on beach erosion on the east coast than the west coast during large storms. While SEPI and water level data do not account for potential wave runup (Stockdon et al., 2007; 2023), Zhang et al. (2001) found a linear relationship between extreme storm surges and storm

waves (wave heights > 2 m) indicating that storm surges make excellent surrogates for storm waves in representing the strength of large storms. The use of storm surge data over wave data is further motivated by the reliability and long-term availability of water level, storm tide, and storm surge data.

In order to contribute to the SEPI, water level data must exceed two thresholds:

1. The verified water level $V$ (the storm tide) must exceed the average Mean High Water, $\overline{MHW}$ (Fig. 1).
2. The storm surge $S = V - P$ (the height of the storm tide $V$ above the predicted tide $P$) must exceed two standard deviations above its average of zero (Fig. 2).

The second criterion is a statistical barrier that only selects the most extreme storm surges during the period of record (Zhang
et al., 2000, 2001). The threshold of storm surges in excess of two standard deviations is consistent with previous research (e.g., Zhang et al., 2000, 2001) that found a linear relationship exists between two standard deviations of the storm surge and large waves ($H_s$ > 2m). Additionally, using two standard deviations provides reasonable results by not identifying too many storms or too few storms relative to named storms (see Results and Supplements 1 and 2).  Given the effect of the water level on storm wave attenuation, these criteria determine the inland extent of storm impact and where on the beach profiles storm
waves will erode or deposit sediment (e.g., Dean, 1991; Edelman, 1972; Steetzel, 1993; Balsillie, 1999; McInnes et al., 2014; Harley, 2017).

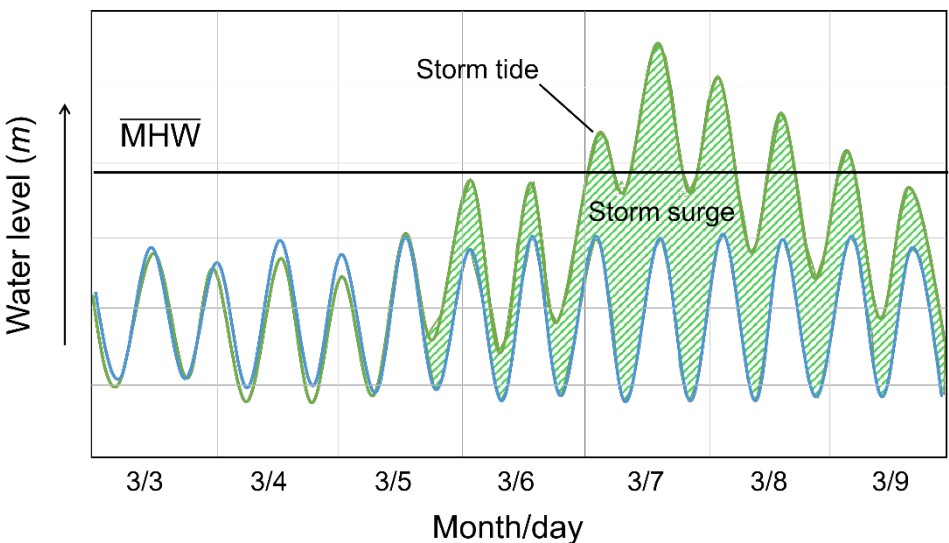

Figure 1: The verified water level (green) and the predicted tide (blue) for a sample storm.  The storm surge (shaded green) is the height of the water level (storm tide) above the predicted tide.

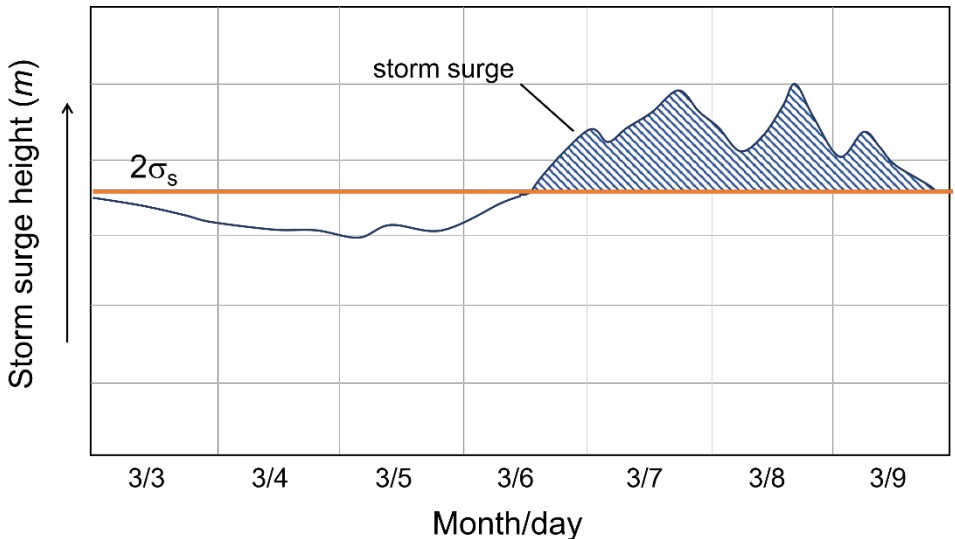

Figure 2: Fictitious storm surge (height of the storm tide above the predicted tide) plotted with the storm surge threshold ($2\sigma_s$, orange), which is two standard deviations above the average surge.

The SEPI magnitude for a single storm is defined in terms of these two thresholds:


$$SEPI = \sum_{t=0}^{t_D} H_{MHW}(t)\, S_{2SD}(t)(\Delta t), \tag{1}$$

where $H_{MHW}(t)$ is the water level (storm tide) above the Mean High Water ($\overline{MHW}$) (Fig. 1),

$$H_{MHW}(t) = \begin{cases} V(t), & V(t) \geq \overline{MHW} \\ 0, & V(t) < \overline{MHW} \end{cases}, \tag{2}$$

$S_{2SD}$ is the storm surge above the threshold of two standard deviations of storm surge, $\sigma_S$ (Fig. 2),

$$S_{2SD}(t) = \begin{cases} S(t), & S(t) \geq 2\sigma_S \\ 0, & S(t) < 2\sigma_S \end{cases}, \tag{3}$$


of a semi-diurnal tide (12 hr; Zhang, 1998). Terms in the sum of Eq. 1 will be zero unless both thresholds are met. Data that exceed both thresholds are grouped together as a single storm (and comprise the terms of the sum in Eq 1) when they are

clustered within 12 hours of each other.  In other words, distinct storms must be separated by 12 or more hours. The duration of the storm is the time difference between the first and last terms of the sum in Eq. 1, and there is no minimum duration required for a storm.

## 2.2 Cumulative storm impact determination

To characterize the cumulative impact of successive storms on sandy beaches, we use the Cumulative Storm Impact Index (CSII) proposed by Fenster and Dominguez (2022). This index accounts for the timing and strengths of previous storms, which make beaches more vulnerable to continued erosion (Fenster and Dominguez, 2022).  The CSII for the $i^{th}$ storm, $I_i$, is equal to the sum of the SEPI of the $i^{th}$ storm, as defined above, and a weighted factor of the previous storm's CSII:

$$I_i = SEPI_i + W_i(t_p)I_{i-1},$$

(4)

where $t_p$ is the time between the $i^{th}$ and the $i^{th}$-1 storm.  Assuming that the recovery rate is proportional to the amount of erosion, we use an exponentially decaying weighting factor for $W_i$ (Fenster and Dominguez, 2022) where:

$$W_i(t_p) = e^{-f(t_p/t_c)},$$

(5)

and $f$ is a constant scaling factor indicating the rate of beach recovery, and $t_c$ is a characteristic time associated with the timescale of beach reset.  More compactly, we write the weighting factor in terms of a rescaled time $\tau_p = t_p/t_c$ and scaling parameter $\delta = e^{-f}$:

$$W_i(\tau) = (\delta)^{\tau^p},$$

(6)

where $\delta < 1$. Note that the CSII has the same units as the SEPI.

While an appropriate value of the characteristic time, $t_c$, is crucial to understanding the meaning of the weighting function, mathematically the two parameters $t_c$ and $\delta$ may be combined into one parameter to achieve the appropriate behavior of CSII (see Fenster and Dominguez (2022) for additional details.)  A reasonable choice of parameters will show accumulation due to storms clustered in time and will show beach recovery (CSII decreasing towards 0) when storms are temporally distant. In practice, there are a range of parameter values that satisfy these conditions and show robust cumulative behavior, though the absolute values of CSII will fluctuate with specific parameter choices.  In this comparative study, we choose a value of $t_c$ = 1 year corresponding to the winter-summer beach profile cycle for beach systems on the U.S. east coast, and $\delta=0.3$ for

consistency across all tidal gauges studied.

The CSII is an impact parameter that also includes the effect of previous storms. For this study, if missing data prevented the analysis from capturing a storm, that storm will not contribute to the CSII of the current storm. Therefore, the CSII is a conservative quantity, which may be higher if more storms were detected.


## 3 Study area and data retrieval

We retrieved water level data from the NOAA Tides and Currents database at 12 tide stations from Portland, Maine to Key West, Florida (tidesandcurrents.noaa.gov; Table 1, Fig. 3). The stations were chosen to obtain the greatest spatial coverage with relatively equal spacings between stations in order to capture variation associated with storm tracks. Additionally, we

sought to acquire data from stations that had the most complete long-term records. Two of the 12 stations are located in areas relatively sheltered from storm conditions and subject to tidal wave amplification. The station at Wilmington, NC (station 8658120) is located on the Cape Fear River approximately 41 km upstream from its connection to the Atlantic Ocean at the Cape Fear River Entrance (Inlet). Similarly, the station at the Battery, NY (station 8518750) is located on the Hudson River,

Table 1: Tide gauge stations along the U.S. East Coast used in this study.

| Station Name | Station ID | Latitude | Longitude | Temporal Data Range |
|---|---|---|---|---|
| Portland, ME | 8418150 | 43° 39.5 N | 70° 14.7 W | 1912-2022 |
| Boston, MA | 8443970 | 42° 21.2 N | 71° 3.0 W | 1921-2022 |
| Newport, RI | 8452660 | 41° 30.3 N | 71° 19.6 W | 1938-2022 |
| Montauk, NY | 8510560 | 41° 2.9 N | 71° 57.6 W | 1947-2022 |
| The Battery, NY | 8518750 | 40° 42.0 N | 74° 0.9 W | 1926-2022 |
| Sandy Hook, NJ | 8531680 | 40° 28.0 N | 74° 0.6 W | 1932-2022 |
| Atlantic City, NJ | 8534720 | 39° 21.4 N | 74° 25.1 W | 1922-1969, 1971-2022 |
| Sewells Point, VA | 8638610 | 36° 56.6 N | 76° 19.7 W | 1927-2022 |
| Wilmington, NC | 8658120 | 34° 13.6 N | 77° 57.2 W | 1936-2022 |
| Charleston, SC | 8665530 | 32° 46.8 N | 79° 55.4 W | 1922-2022 |
| Fernandina Beach, FL | 8720030 | 30° 40.3 N | 81° 28.0 W | 1938-2022 |
| Key West, FL | 8724580 | 24° 33.0 N | 81° 48.5 W | 1926-2022 |

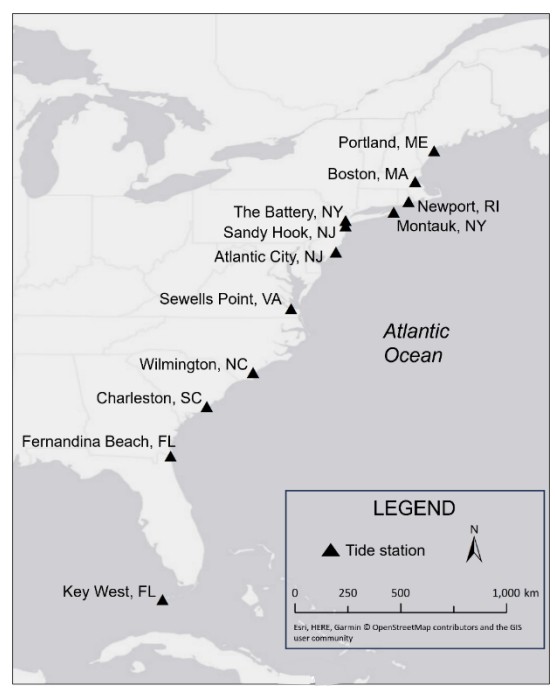

Figure 3: Locations of selected tidal gauges along the U.S. east coast corresponding to Table 1.

approximately 15.5 km upstream of its mouth at Lower Bay, New York Harbor. It should be noted that these two stations are most likely subject to tidal wave transformation caused by interactions with complex channel geometries of shallow estuaries and/or fluvial processes not prone to occur at stations located along the open ocean coast (e.g., Aubrey and Speer, 1985; Speer and Aubrey, 1985; van Rijn, 2011; Hein et al., 2021). We take these conditions into account when drawing inferences from our results below.


For each station, we retrieved the following data from NOAA (McManus et. al., 2023a, 2023b; Table 1):

- $V(t)$: Hourly verified water levels measured relative to the station datum (STND)
- $P_{CTE}(t)$: Hourly predicted tidal levels relative to STND, and centered on the mean sea level (MSL) tidal datum of the current tidal epoch (CTE), in order to calculate actual predicted tidal levels
- $MHW_{MM}$: The monthly means of the mean high water (MHW) in order to calculate annual averages
- $MSL_{MM}$: The monthly means of the MSL datums in order to calculate annual averages
- $MSL_{CTE}$: The MSL for the current tidal epoch (1983-2001)

We note that Zhang et al. (2000, 2001) relied on the condition that water levels exceed the mean higher high water (MHHW)
value, rather than the MHW, to identify storms.  This is because storm tide above MHHW is high enough to directly and

forcefully attack the dunes (Zhang, 2001). Because the U.S. east coast experiences largely semi-diurnal tides and because the difference between MHW and MHHW is small, it was not standard practice for NOAA's National Ocean Service to calculate historical MHHW values at tide gauges located on the U.S. east coast (T. Ehret, NOAA, personal communication). Furthermore, we conducted sensitivity analyses to determine the differences between MHW and MHHW for more recent years when water level data from both datums were available. These analyses revealed no significant differences in storm identification results for the stations considered using MHW compared to MHHW. It should be noted, however, that MHW should not replace the MHHW threshold in general. It is not expected that the MHW level is high enough for waves to do significant work on dunes for mixed semi-diurnal systems such as the U.S. West Coast.

Because the SEPI relies on a water level (in this case MHW) to identify storms, we had to remove the trend bias associated with relative sea-level rise that has occurred ubiquitously along the U.S. east coast since the beginning of the tide station data used in this study (1912; Sweet et al., 2017). To accomplish this task, we averaged the available mean monthly $MHW_{MM}$ values (daily values are not available) to calculate the average annual mean high water value, $\overline{MHW}$, to obtain the first (water level) threshold as discussed above (see Fig. 4 for an example at Sewells Point, Virginia, tide gauge station 8638610).

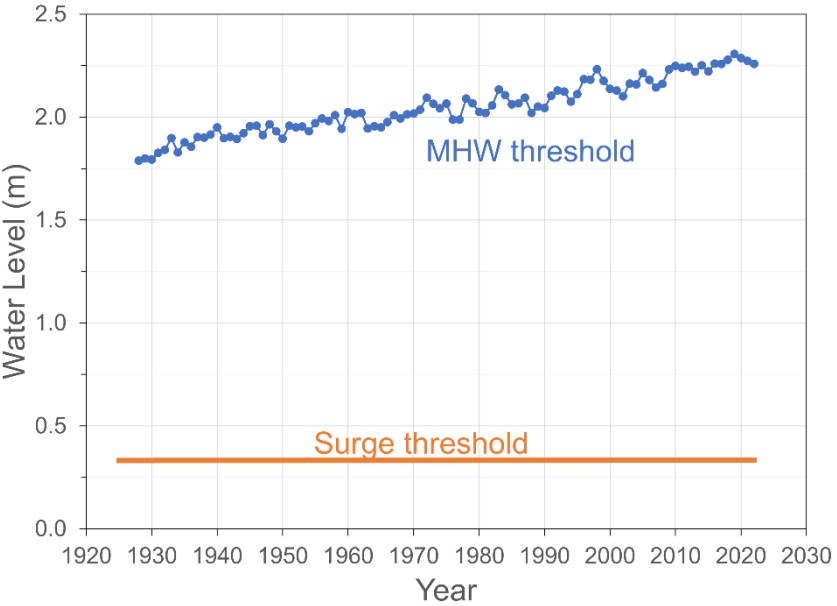

Figure 4: Example of two exceedance thresholds required for water level data to contribute to the SEPI storm magnitude at Sewells Point, Virginia (tide gauge station 8638610). Water levels must exceed the Mean High Water (MWH) and the storm surge must exceed two standard deviations above zero. The MWH threshold is calculated annually to account for sea level rise, while the surge threshold is a single value calculated from all available data.

To produce the P<sub>CTE</sub>(t) values, NOAA uses a Least Squares Harmonic Analysis at each station to produce a set of *harmonic constants,* which empirically weight 37 different harmonic constituents to account for various astronomical, hydrodynamical, and seasonal influences on the tides for each station (NOAA, 2023; Parker, 2007). NOAA uses the most current set of harmonic constants to generate these predicted water levels $P_{CTE}(t)$ over the entire period of the data retrieval request and sets the mean of *all* predicted levels to be the MSL for the current 1983-2001 tidal epoch (CTE) (T. Ehret, NOAA, personal communication). Tides are expected to oscillate about the mean sea level, but NOAA's method of prediction does not account for the significant sea-level rise that occurs over the time scale of interest. Consequently, the mean of the $P_{CTE}(t)$ values does not match the actual mean sea level for (at least) the time periods prior to the current tidal epoch. We correct for this by setting the mean of the predicted water levels to be equal to the actual mean sea level on any given year. To do this, we retrieved $MSL_{MM}$ sea levels from NOAA to calculate annual averages of the Mean Sea Level, $\overline{MSL}$. We then used the MSL for the current tidal epoch, $MSL_{CTE}$, to calculate the corrected predicted tidal levels as:

$$P(t) = P_{CTE}(t) - (MSL_{CTE} - \overline{MSL}). \tag{7}$$

The hourly storm surge is the measured water level height above the predicted tidal height:

$$S(t) = V(t) - P(t). \tag{8}$$

For all 12 stations, the hourly storm surge data are approximately Gaussian distributed, centered about a value of zero ($\bar{S}$=0). The standard deviation of the hourly surge values, $\sigma_s$, measured over approximately 10 year time intervals, do not change appreciably over time for almost all stations with values ranging from $\sigma_s = 0.07$ m (Key West) to $\sigma_s = 0.17$ m (The Battery and Sandy Hook). The exception is Wilmington, NC, which had a decreasing value from about $\sigma_S = 0.22$ to $\sigma_S = 0.14$ over the time period from 1935-1970. However, a sensitivity analysis revealed no significant differences in storm identification results when an annually calculated storm surge threshold was used instead of a single storm surge threshold. Because storm surge distributions remain approximately constant in time, (in contrast to the MHW values, which rise with MSL and therefore require annually calculated thresholds), we use a single value calculated from the entire data set for each station (cf., Fig. 4).

## 4 Results

We present below the annual trends in storm frequency (number and average annual) and magnitude at all 12 tide gauge stations used to characterize storminess along the U.S. east coast using the SEPI criteria as defined above. We then compare

the cumulative impacts of successive storms and large magnitude storms (CSII) on the potential beach erosion to the SEPI magnitudes of each storm. All storms – tropical and extratropical in origin – contribute to the results at each station. Because some stations became nonfunctional during storm events, we excluded years for which $\geq$ 10% of the water level data was missing to eliminate selection bias, i.e., generating annual averages with missing water level data.

### 4.1 Storm frequency: annual trends

Results from an ordinary least squares regression analysis of storm frequency show no appreciable increase in the number of storms per year at any of the stations except for Wilmington, NC (station 8658120). Testing the null hypothesis that the trend coefficient = 0, we found that only Wilmington, NC emerged as statistically significantly different than 0 (p = 0.001) with an increase in 0.05 storms per year, or one additional storm every 20 years, between 1936-2022 (Figs. 5a, 6a).

Annual average storm counts range spatially from a minimum of 5 storms $\pm$ 4 storms per year at Wilmington, NC (station 8658120) to a maximum of 20 storms $\pm$ 6 storms per year at Newport RI (station 8452660) and Montauk, NY (station 8510560) (Fig. 6a). However, the sheltered location of the Wilmington station may account for this low value. Geographically, the annual storm counts increase from Portland, ME to Newport and decrease from Newport to Sandy Hook, NJ (Fig. 6a). The high temporal variability in annual storm counts (coefficient of variation ranges from 20%-70% and a median of 30%) reflects the large interannual variation in storminess at each location.

### 4.2 Storm magnitude: annual trends

Figure 5b shows the SEPI averaged over all storms each year for each tidal station. Unlike storm frequency, the linear regression significance test (with Bonferroni correction) of change in annual SEPI values over time at each station showed statistically significant increases in time for eight of the 12 tidal stations (Table 2; Fig. 5b). Stations that showed significant increases in magnitude included Montauk, NY, the Battery, NY, and all six stations from Atlantic City, NJ southward to Key West, FL. However, the increase in storm magnitude is modest. Sewells Point, VA, had the greatest increase of 0.080 $m^2$ h/yr. To give some scale, Sewells Point has an overall average storm SEPI of 14 $m^2$ h/yr. It would therefore take 175 years for the average SEPI value to double with the calculated rate of increase and 50% of the data have a SEPI magnitude of $\leq$5 $m^2$h/yr. Similar results are found when a low pass filter of four years is applied to the data in Figure 5b (Supplement 3).

Table 2: Fit parameters for average annual SEPI per storm, corresponding to Fig. 5b. Slopes in boldface are statistically significant at $p \leq 0.05$.

| Station | Slope (m²h/yr) | p value | R² value |
|---|---|---|---|
| Portland | 0.015 | 0.082 | 0.03 |
| Boston | 0.017 | 0.055 | 0.04 |
| Newport | 0.004 | 0.363 | 0.01 |
| Montauk | **0.035** | 0.049 | 0.07 |
| The Battery | **0.045** | 0.002 | 0.13 |
| Sandy H. | 0.028 | 0.063 | 0.04 |
| Atlantic C. | **0.033** | 0.016 | 0.06 |
| Sewell's P. | **0.080** | 0.001 | 0.11 |
| Wilmington | **0.058** | 0.023 | 0.07 |
| Charleston | **0.023** | <0.001 | 0.15 |
| Fernand. B. | **0.020** | 0.019 | 0.07 |
| Key West | **0.028** | 0.001 | 0.11 |

355

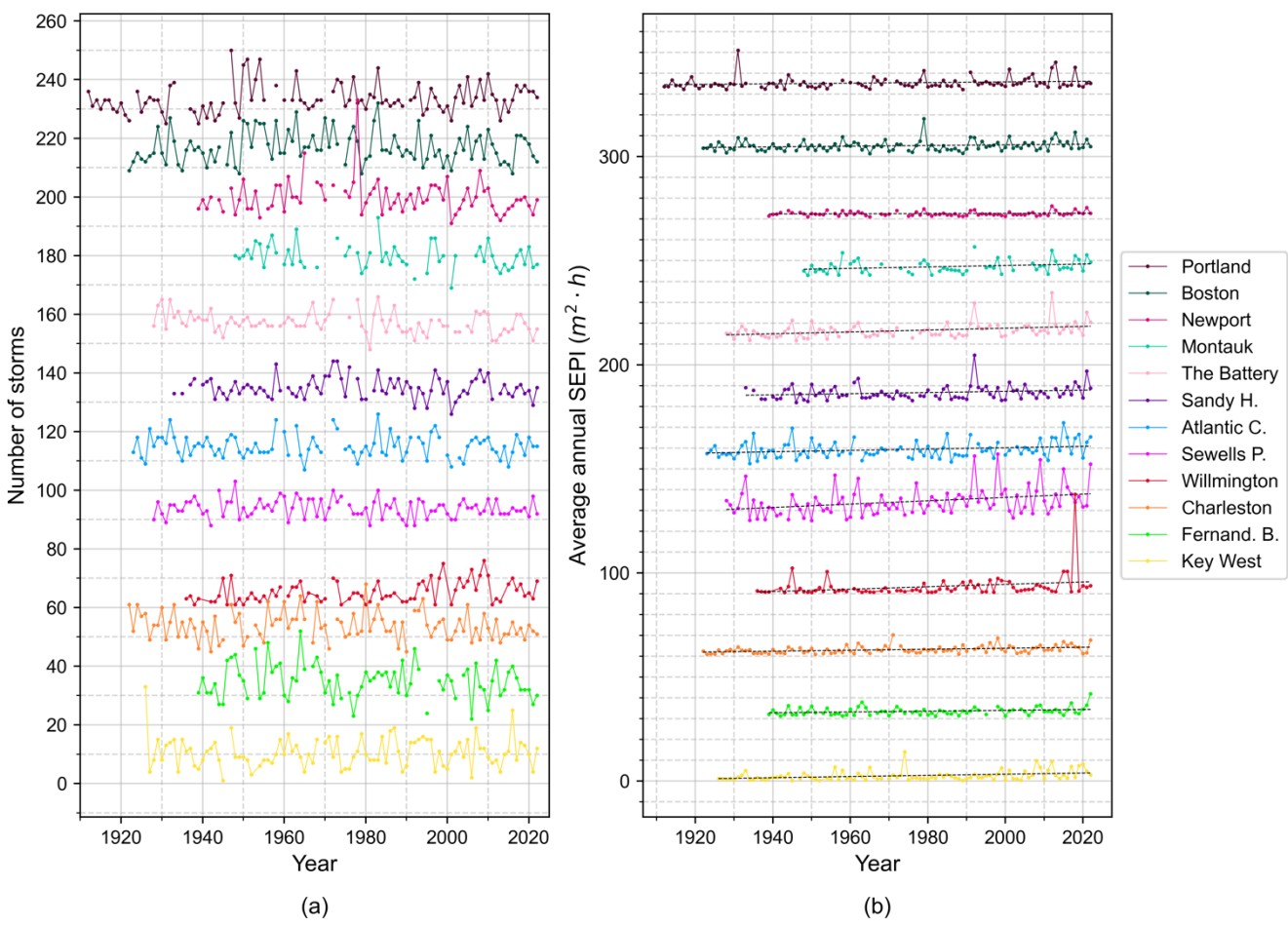

Figure 5: Annual results of (a) storm frequency (number of storms/year) and (b) storm magnitude (SEPI averaged over all storms identified in each year) arranged from north (top) to south (bottom). For clarity, the frequency data series (a) from each station is offset vertically by 20 storms. A linear regression trend analysis showed no appreciable increase of the frequency of storms over the period of record. For clarity, storm magnitude data sets (b) are offset vertically by 30 $m^2$ h. Unlike frequency, a statistically significant increase in values exists for most stations (8 out of 12). For both (a) and (b), years missing 10% or more data were not plotted. Consecutive years of valid data are connected by lines.

The average annual SEPI values also vary geographically (Fig. 6b). The values increase from Newport, RI (north) to Sewells Point, VA (south) (except for Montauk) and show an opposite trend to the storm frequency, i.e., average annual storm magnitudes increase while storm frequencies decrease from Rhode Island to Virginia. The lowest average annual storm magnitudes come from Newport, RI and the four southernmost tide stations from Wilmington, NC to Key West, FL.

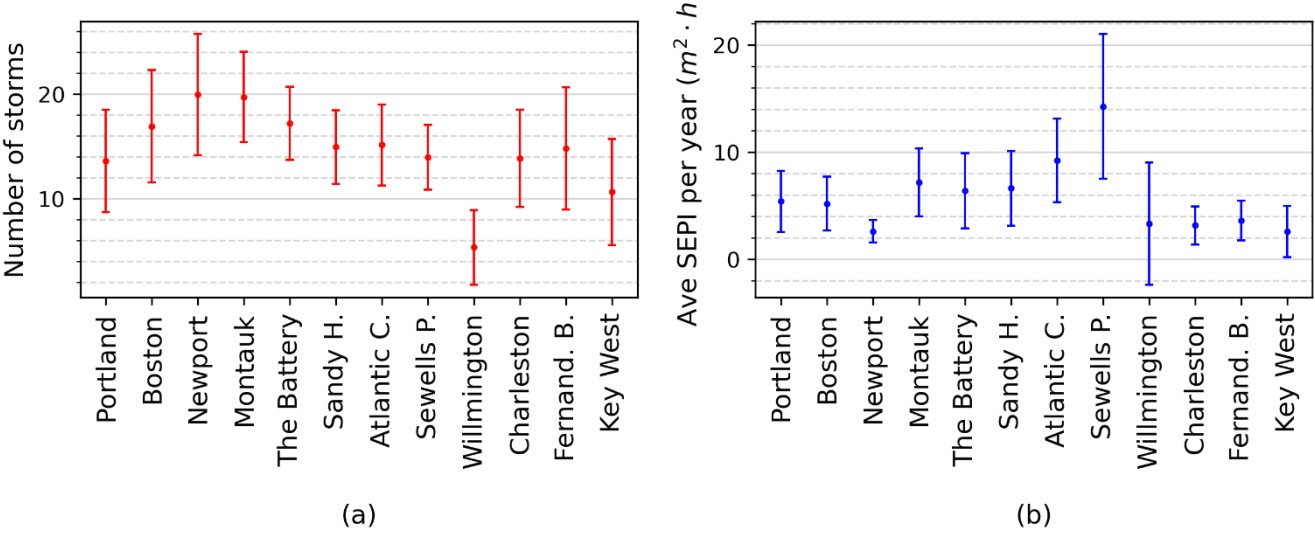

 Figure 6: Average and standard deviation of each data set in Fig. 5. Calculations include all years of data plotted in Fig. 5, and similarly exclude years for which ≥10% of data are missing.

## 4.3 Cumulative impact of storm clustering (frequency) and magnitude

Plots of the individual storm SEPI magnitudes at all tide stations over the entire time record used in this analysis show similar results as the average annual frequency and magnitude results, i.e., no trend in frequency but larger magnitude "SEPI storms" apparent in the more recent decades (Fig. 7). Plots of CSII values using δ=0.3 at each station show the effect of the larger storms and storm clustering on the beach erosion potential (Fig. 8; Fenster and Dominguez, 2022). Peaks in the CSII values correspond to time periods when beaches were most vulnerable to erosion (least recovered) and the receding limb following the peak corresponds to the beach recovery phase (or, in some cases, missing data; Fenster and Dominguez, 2022). The results plotted in Fig. 8 also show that the tide stations located geographically adjacent to each other from Montauk, NY south to Sewells Point, VA have the greatest magnitude CSII values, more frequent peaks, and longest recovery times (≈4 years on average) of tide stations to the north and south of this region. In addition to the larger magnitude CSII values, these four stations show an increasing trend in CSII magnitudes beginning c. 1990s – 2000s. Finally, the CSII peaks appear to have a periodicity on the order of 3-10 years for all stations except for Newport, RI and Wilmington, NC and spatial autocorrelation corresponding with storm track and intensity during the time period in which the storm reached a particular tide gauge (Fig. 8).

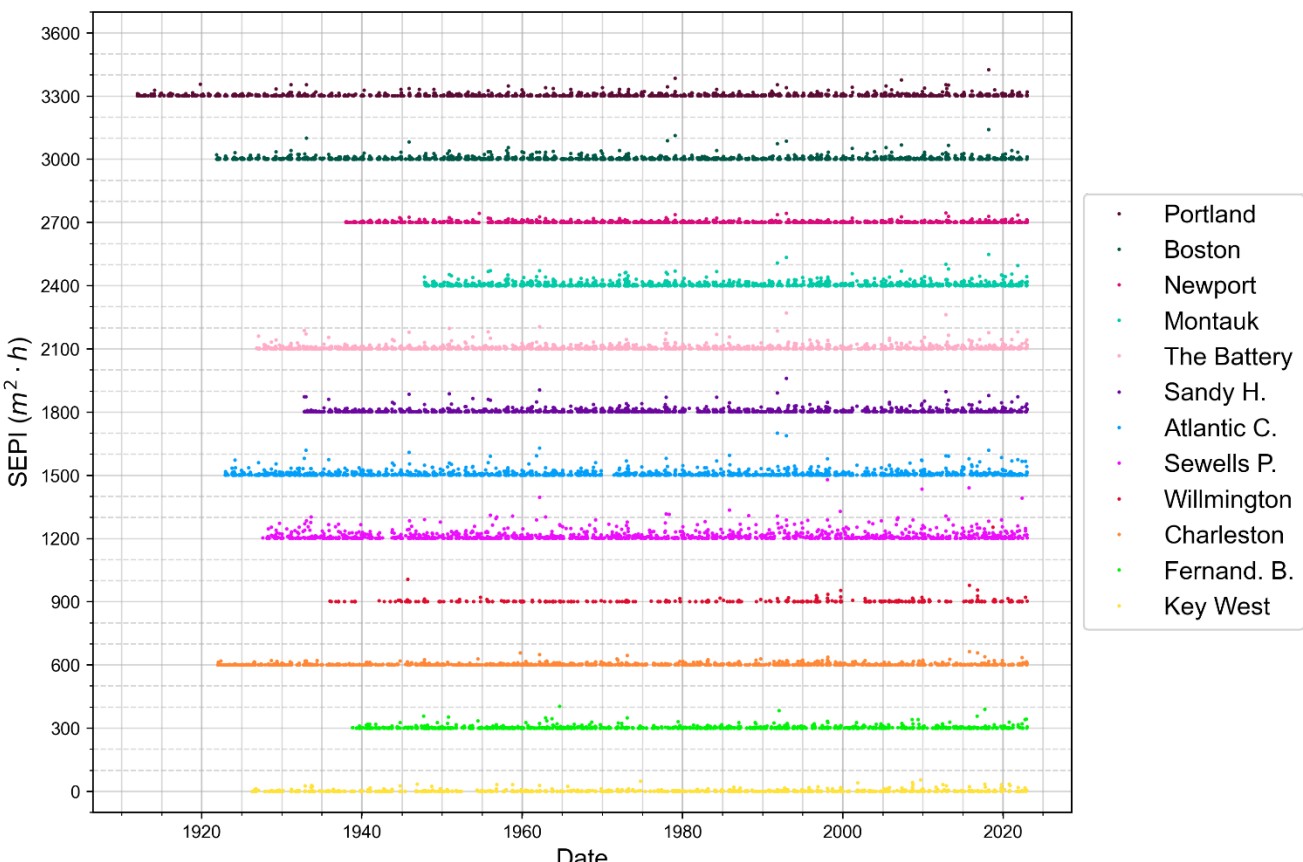

Figure 7: SEPI values for each storm identified over the periods of record for all 12 tide gauges along the U.S. east coast. For clarity, datasets are offset vertically by 300 $m^2$ h. Note data sets are arranged from north to south and use the same color scheme as Fig. 5.

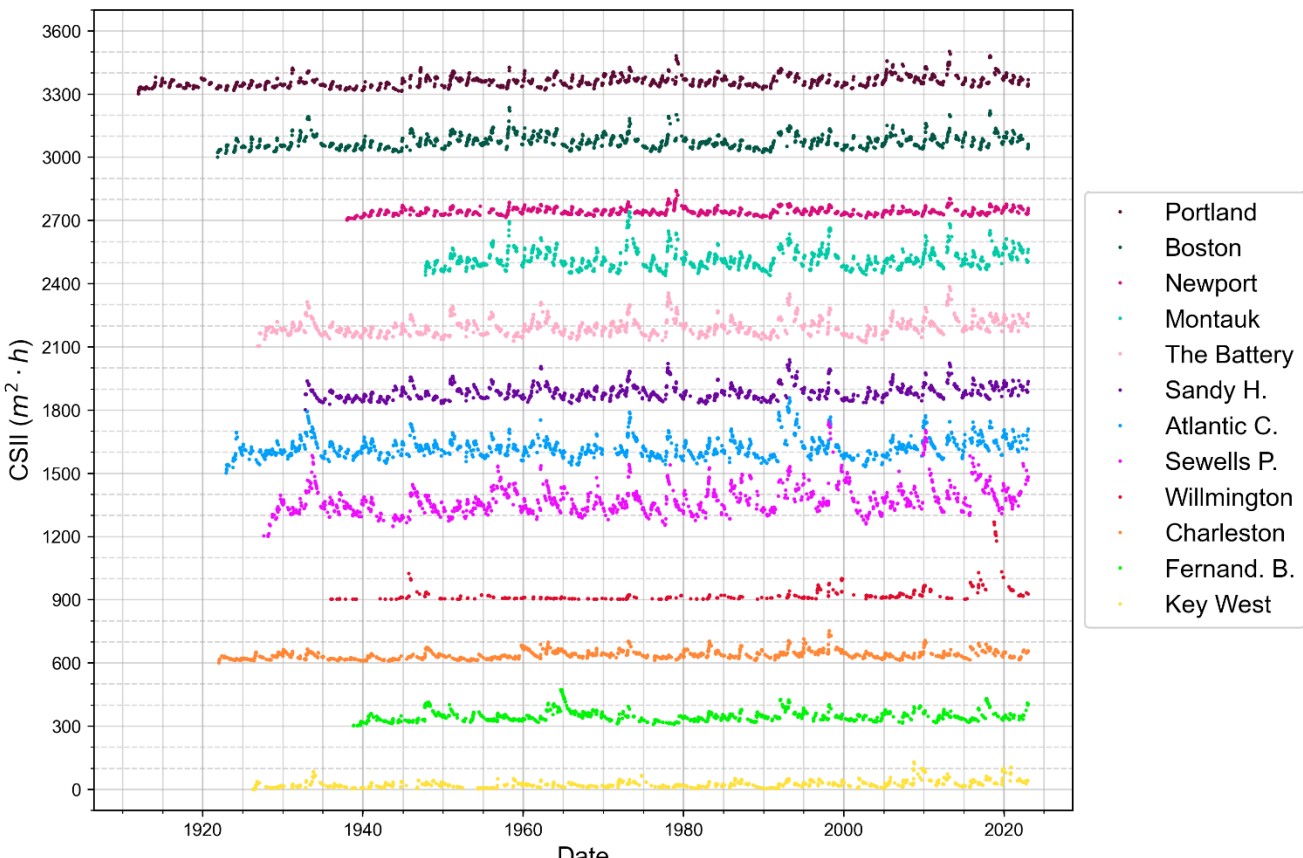

Figure 8: CSII ($\delta = 0.3$) values for each storm identified over the periods of record for all 12 tide gauges along the U.S. east coast. For clarity, datasets are offset vertically by 300 $m^2$ h. Note data sets are arranged from north to south and use the same color scheme as Fig. 5.

For comparison, we plotted both the SEPI and CSII values for two sample stations: Newport, RI in the north and Fernandina Beach in the south (Fig. 9). The SEPI values in Newport station correspond to the seasonal extratropical storm cycle and the Fernandina Beach station reflect tropical storm seasonality. Additionally, the large CSII values indicate that successive and large magnitude storms accumulate beach erosion potential and do not allow full beach recovery thereby masking the seasonal cyclonic variation (Fig. 9). The time spans associated with beach recovery range from ≈3 years to >10 years depending on the storminess in time and space.

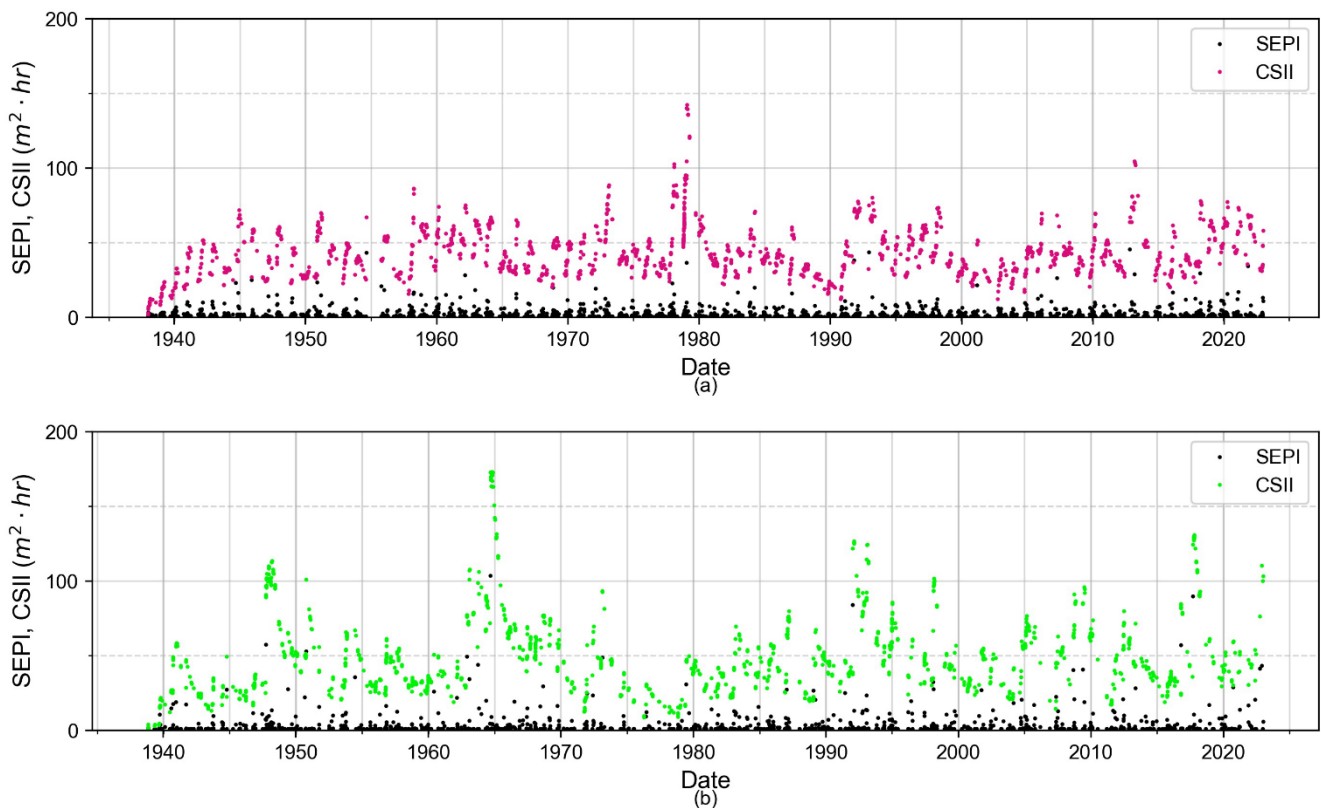

Figure 9: Comparison of SEPI and CSII measures of storm magnitude for two selected tide gauges: (a) Newport, RI (to the north) and Fernandina Beach (to the south). The SEPI (lower value) is plotted in black and CSII in color.

The effect of individual named storms on the CSII values over a sample of recent time (2008-2022) shows regional (spatial) correlations in the cumulative effect of beach erosion potential (Fig. 10). The CSII peaks in the northern tide stations correspond to extratropical storms (Fig. 10) and the CSII peaks in the southern tide stations correspond to tropical storms (Fig. 11) thus emphasizing the importance of both storm clustering and magnitude on the cumulative beach erosion potential: More frequent and larger named storms lead to large accumulation of beach erosion potential and conversely, the lack of

named storms can lead to beach recovery (Fig. 10). These phenomena can occur regionally and over varying time scales. For example, the U.S. east coast experienced six notable nor'easters during 2009-2010 (including Nor'ida), five during the 2012-2013 storm season (including "super storm Sandy"), and six in the winter and spring of 2017-2018 (Fig. 10., Supplement 1). The large CSII peak during the 2018 tropical storm season observed for the Wilmington, NC tide station corresponds to storm tracks associated with tropical storm Beryl and hurricane Chris. The two named storms occurred within a week of each

other, and the latter stalled and intensified from a tropical storm to a category 2 hurricane off the North Carolina coast. Later that season, the category 1 hurricane Florence made landfall at the North Carolina-South Carolina border placing the

Wilmington station in quadrant 1 of this storm. The tide stations to the south of Wilmington did not record these storms appreciably. Despite the particularly stormy July 2018 along the North Carolina coast, we note that the peak stream stage measured at the stream gauge in closest proximity to (and upstream from) the Wilmington tide gauge (Cape Fear R at Lock 1

NR, Kelly, NC, 02105769) was approximately 11.6% greater than the mean stage (5.4 m vs. 4.9 m, respectively) and 15.5% greater than the median stage (5.4 m vs. 4.7 m, respectively). Consequently, post-storm river flow most certainly impacted the Wilmington water level data by approximately 11-16%. However, several tropical storms occurring during summer/early fall 2016 (i.e., Bonnie, Colin, "Eight 2016", Hermine, Julia, and category 2 hurricane Matthew) did have a regional impact that included Wilmington, NC south to Fernandina Beach, FL (Fig. 11; Supplement 2). Finally, these results

show that unnamed, smaller magnitude extratropical storms clustered in time can impact CSII without the effects of a named storm (c. 2018-2021, Fig. 10, cf. Fenster and Dominguez, 2022).

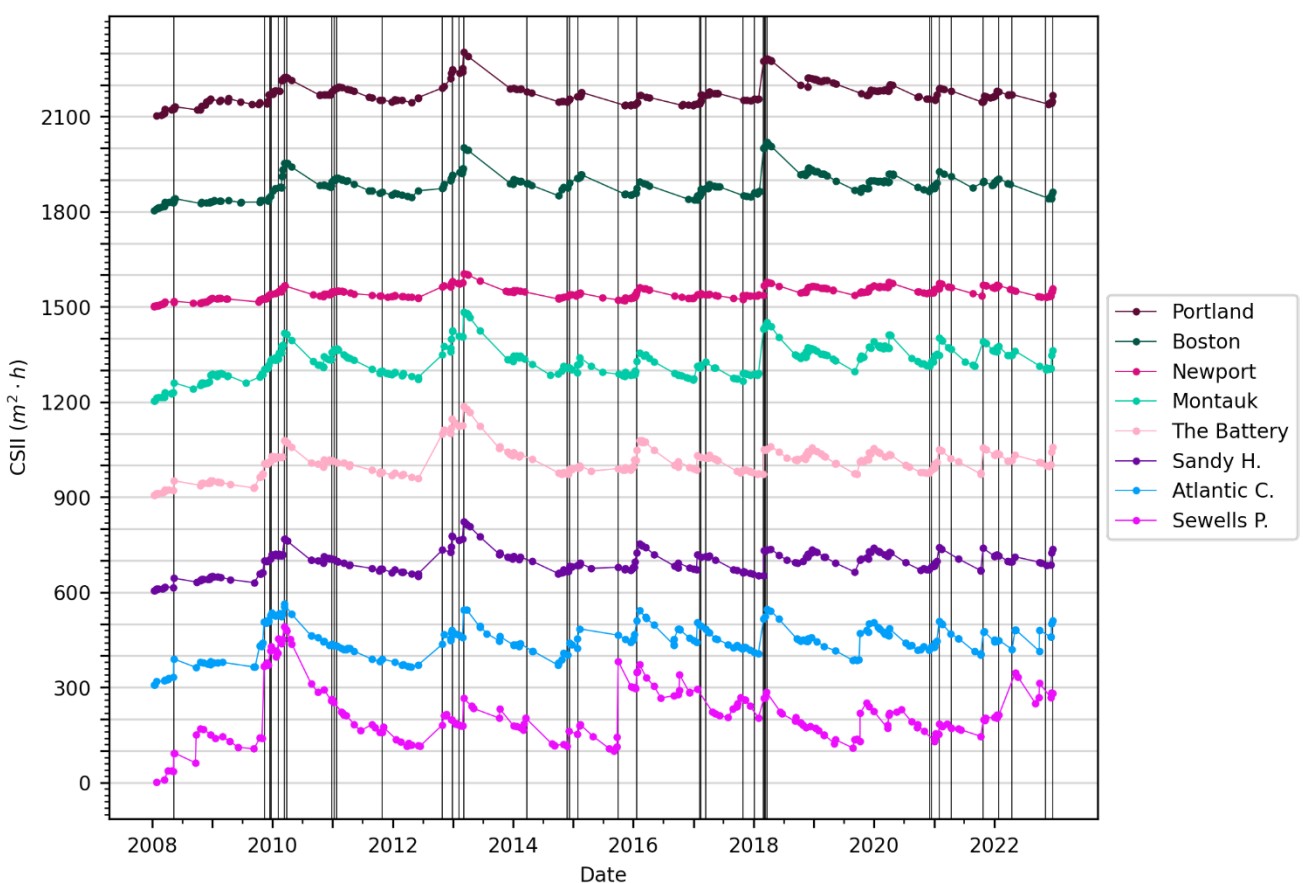

Figure 10: CSII values plotted for the period 2008-2022 for all stations used in this analysis with black vertical lines marking
the date of major (named) extratropical storms for **northern tide stations** (Portland ME to Sewells Point, VA). For clarity,

datasets are offset vertically by 300 m$^2$ h, as in Fig. 7 and Fig. 8. Note that greater storm clustering of named storms corresponds with greater density of vertical lines.  Supplement 1 contains a table of the named extratropical storms shown on these plots.

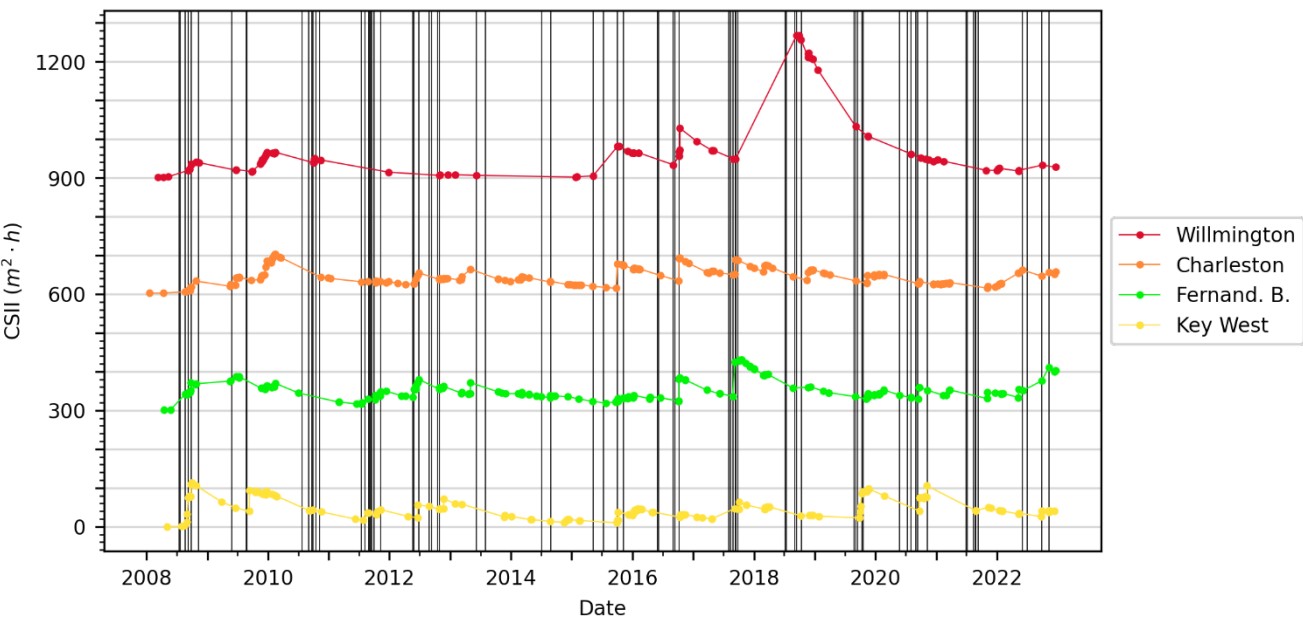

Figure 11: CSII values plotted for the period 2008-2022 for all stations used in this analysis with black vertical lines marking the date of major (named) tropical storms **for southern tide stations** (Wilmington, NC to Key West, FL). For clarity, datasets are offset vertically by 300 m$^2$ h, as in Fig. 7 and Fig. 8.  Note that greater storm clustering of named storms corresponds with greater density of vertical lines.  Supplement 2 contains a table of the named tropical storms shown on these plots.

**5 Discussion**

This study extracted historical water level data from 12 NOAA tide gauge stations, spanning the early 20th century to 2022 from central Maine to southern Florida, to determine if temporal and spatial trends existed in storm  frequency and magnitude along the U.S. Atlantic Ocean coast. We used the Storm Erosion Potential Index (SEPI) to provide thresholds for storm surges and tides that defined a storm by extreme water levels  (Zhang, 1998; Zhang et al., 2000, 2001). Additionally,
we examined the potential beach erosion impacts of successive and large magnitude storms by incorporating SEPI magnitudes into a newly developed cumulative storm impact empirical model (Fenster and Dominguez, 2022).

Our methods avoid typical problems associated with empirical data analyses such as using heterogeneous or hindcasted historical instrumental data and limited temporal data to detect long-term (decadal to centennial) trends in cyclone frequency or intensity (e.g., Schreck et al., 2014; Seneviratne et al., 2021). Consequently, our results (using the SEPI definition) can be

used to validate global and regional modelling studies of storm frequency in the Atlantic basin which show no significant increasing or decreasing trend in storminess (Fig. 5a; Colle et al., 2015; Tory et al., 2013; Knutson et al., 2020; Seneviratne et al., 2021; Sobel et al., 2021).  Average annual storm counts ranged from five to 20 at the 12 tide stations of interest, but the annual storm frequency did not change over time significantly at any of these stations, except for a minor increase at the Wilmington, NC tide station. This result also builds on the conclusions of Zhang et al. (2000) whose study, which ended in 1996/97, found no discernible long-term trend in the historical number of storms occurring along the U.S. east coast using the SEPI criterion through 1997.

With respect to average annual storm magnitudes, we found statistically significant, but modest increases through time for eight of the 12 tidal stations (Table 2; Fig. 5b). Of the statistically significant stations, slopes of SEPI magnitudes per year ranged from 0.02 additional storms per year to 0.08 additional storms per year and the statistically significant increases all occurred in the stations from New York to Florida (excluding Sandy Hook, NJ; Table 2; Fig. 5b).  This finding is consistent with that of Tadesse et al. (2022) who showed significant increases in the frequency of extreme storm surges over the 95th percentile (2 $\sigma$) and 99th percentile (3 $\sigma$) along the New England (northeast U.S.) coast using storm surge time series data from the Global Storm Surge Reconstructions (GSSR) database. Tadesse et al. (2022) do not identify individual storm events, but rather count of the number of daily storm surge exceedances over various thresholds. Therefore, their calculation of positive frequency trends is more akin to our magnitude trends for the average annual SEPI (Figure 5b). The modest increase in the storm magnitude at the eight stations to the south of New England suggests that the well-documented increase in shoreline damage (Smith and Katz, 2013; Smith and Matthews, 2015; Harley, 2017; NOAA National Center for Environmental Information, 2024; Callahan et al., 2022) is most likely due to other (or a combination of) factors such as an increase in frequency and/or intensity the largest storms (e.g., Komar and Allan, 2008; Walsh et al., 2016; Knutson et al., 2020; Kossin et al., 2020; Ghanavati  et al., 2023) and/or sea-level rise (FitzGerald et al., 2008): the same number of storms will do more overall damage when the work done on beaches is located higher up on the beach profile (Sallenger, 2000; Stockdon et al., 2007; Birchler et al., 2015) and/or when deficits in coastal sediment budgets exist (Ghanavati et al., 2023).

The reason(s) the SEPI magnitudes have increased modestly at the southern stations, but not at the northern stations (from Rhode Island to Maine) may include (1) orogenesis; extratropical cyclones that originate over Florida or north of Cuba, travel northward, and increase in magnitude when they become blocked by stagnating anticyclones over the North Atlantic or New England (Davis et al., 1993); (2) sea surface temperatures; tropical storms that originate in the Atlantic basin are correlated with sea surface temperatures which have increased in the time span of our data and analyses (Kossin et al., 2017; Knutson et al., 2020) , but also tend to weaken and veer eastward (away from the coast) in the westerly global atmospheric currents before reaching the New England states; and/or (3) exposure; high-latitude stations experience more consistent large-scale extratropical cyclones associated with the northerly positioning of the jet stream during winter and early spring months (Davis et al., 1993). The latter explanation (3) is supported by the finding that the average SEPI per year is generally

smaller for the four southernmost tide stations, but the number of storms per year is larger for the northernmost tide stations (Figs. 5. 6, 7a). However, our findings do not corroborate northward shifting of tropical cyclone storm tracks (Kossin, 2018; Knutson et al., 2019; Murakami et al., 2020; Yang et al., 2020).

Our results showed that storms become larger and less frequent from Rhode Island to Virginia indicating that the mid-Atlantic is more prone to larger storms than the northerly tide stations. Sewells Point, VA had the largest absolute average annual SEPI magnitude and the largest increase in SEPI magnitudes over time. The increase in SEPI magnitude at this location was more than double that of the magnitude increase at all other statistically significant stations except for Wilmington, NC and the Battery, NY which both reside in sheltered tidally influenced rivers and experience augmentation of
the tidal wave/surge and/or impacts by fluvial processes (Aubrey and Speer, 1985; Speer and Aubrey, 1985; Van Rijn, 2011; Hein et al., 2021). This finding is to be expected given its geographic location in the middle of the Atlantic coast and consequent exposure to tropical, extratropical and transitional storms; and the station location relative to the unlimited east fetch of the Atlantic Ocean and the north-facing maximum fetch of the Chesapeake Bay (Nadal-Caraballo et al., 2015; Callahan et al., 2022).

The new CSII parameter gives a perspective of storm impact by using any storm metric to account for the magnitude of an individual storm and the timing and temporal clustering of storms (Fenster and Dominguez, 2022).  For this study, we selected the SEPI as a metric to assess spatial and temporal trends in storminess and potential beach erosion (Zhang et al., 2000, 2001). When CSII values approach SEPI values, the beach profile is in a more accretionary (or equilibrium) state and when the CSII values become much larger than the SEPI values, the beach is in its most vulnerable erosional state which
becomes the antecedent condition for the impact of the next sequential storm (Fig. 12). Accretionary states occur during periods of storm quiescence and erosional states occur following storm clustering in time or large magnitude storms (assuming no changes to the sediment supply and fluxes). The peak CSII values indicate the time at which beaches are in their most vulnerable state and the "troughs" suggest the beach has approached recovery. The results from this study show that peaks and troughs tend to vary on time scales of four to 10 years and provide insight into the time scale allowed for
beaches to "heal" after storm clusters and large magnitude storms occur (Figs. 8, 9, 10, and 11). This four-to-10-year time period varies temporally and spatially with some locations apparently more periodic in nature (e.g., Sewells Point), some places less periodic (e.g., Wilmington), and other locations more periodic in recent times (e.g., Charleston). While causative explanations for these variations are beyond the scope of this study,  the aperiodic clusters have been thought to correspond with interdecadal to decadal scale variability observed in cyclonic development caused by North Atlantic Oscillation (NAO)
and El Niño Southern Oscillation (ENSO) phases (Figs8, 9, 10, and 11; Davis et al., 1993; Zhang et al, 2000; Hirsch et al., 2001; Colle et al., 2015).

Finally, this study identified the well-known latitudinal gradient between tropical and extratropical storms along the U.S. east coast by revealing the tendency for tropical storms to control beach erosion potential along southern U.S. coasts (Wilmington, NC to Key West, FL) and extratropical storms to impact the northern beaches (Portland, ME to Sewells Point,

VA) (Fig. 10 and Fig. 11; Davis et al, 1993; Zhang et al., 2001; Booth et al., 2016). As noted above, Sewells Point, VA contained the greatest magnitude SEPI and CSII values because of its location in the mid-Atlantic reach and exposure to both storm types. Additionally, CSII peak magnitudes were greater for the northern, extratropical storm influenced stations than the CSII peak magnitudes of the southern stations (except for Wilmington, NC given the potential augmentation of the surge as discussed above) demonstrating the importance of more frequent, extratropical storms in affecting beach states in

the north. However, we would expect future CSII values to increase in the southern stations given the potential for sequential tropical cyclone impacts to increase in the future (Xi et al., 2023).

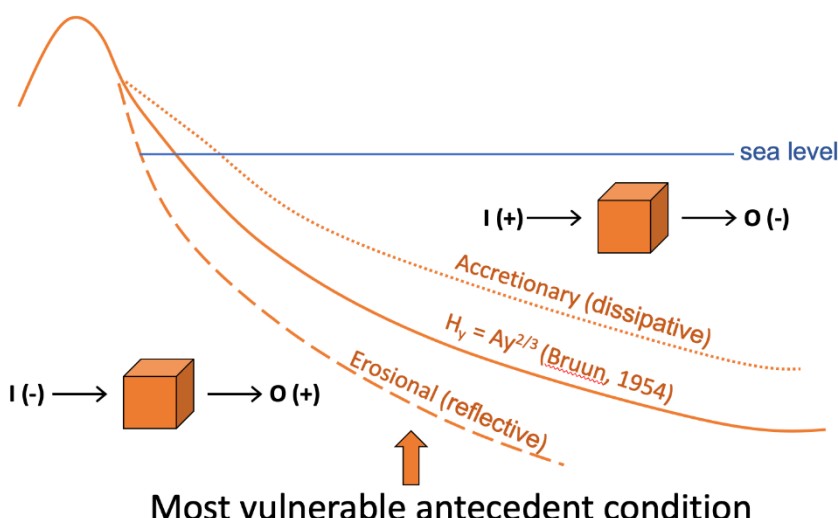

Figure 12: Various antecedent beach states that precede a storm impact. The cubes represent control volumes into which an input (I) of sand transports and out of which an output (O) of sand leaves. Erosional states reflect I < O and accretionary states

indicate I > O.

## 6 Conclusions

This research provides a comprehensive analysis of storm frequency and magnitude trends along the U.S. east coast, ranging from central Maine to southern Florida. Using historical water level data from 12 NOAA tide gauge stations and the Storm

Erosion Potential Index (SEPI) criteria to identify storm events, this study examines both temporal and spatial variations in storminess and its potential impact on beach erosion.

The analysis of storm frequency reveals no significant long-term trend across the studied stations, except for a minor increase observed at the Wilmington, NC tide station. However, the examination of storm magnitude indicates statistically

significant but modest increases over time at eight out of the 12 tidal stations, particularly evident from New York to Florida. These findings align with previous studies indicating an increase in extreme storm surges along the U.S. east coast, suggesting potential implications for shoreline damage.

Moreover, the research introduces a novel metric, the Cumulative Storm Impact Index (CSII), which accounts for both storm

magnitude and clustering in time. The CSII analysis highlights the cumulative impact of successive and large magnitude storms on beach erosion potential, revealing distinct regional patterns. Notably, the study identifies a latitudinal gradient in storm impact, with tropical storms predominantly influencing southern U.S. coasts and extratropical storms impacting northern beaches.

Overall, this research enhances our understanding of storm dynamics and their implications for coastal erosion along the U.S. east coast. By providing insights into both frequency and magnitude trends, as well as the cumulative impact of storms, the findings contribute valuable information for coastal management and resilience planning in the face of changing climatic conditions. Further research in this area is crucial for anticipating future storm impacts and implementing effective mitigation strategies to safeguard vulnerable coastal communities and ecosystems. Such studies should include identifying

regional and local factors that control the beach recovery time and compared that to the time allowed for beach recovery based on the CSII analysis and the observed four-to-10-year interdecadal variation in the observed CSII values.

**Author contributions**

MF and RD conceptualized the project and developed the methodology.  JM curated the data; MF and RD conducted the investigation and formal analysis.  MF and RD wrote the manuscript draft, and RD, MF, and JM reviewed and edited the

manuscript.

**Code and data availability**

The data used for this study come from NOAA Tides and Currents and NOAA About Harmonic Constituents. Software codes used for data curation can be found in McManus et al. (2023a and 2023b). McManus et al. (2023c) provide the codes used to

identify storms and perform the SEPI calculations. Resulting data can be found in McManus et al. (2024d). Dominguez et al.
(2023) contain the code used for CSII calculations and figure creation.

**Competing interests**

The authors declare that they have no conflict of interest.

**Acknowledgements**

This project was funded by, in part, by Randolph-Macon College's Walter Williams Craigie Endowment, Chenery Research Professorship Grant, and Rashkind Endowment. Additional support came from the Virginia Coast Reserve's Long-term Ecological Research NSF grant DEB-1832221. We thank Charles Gowan for his statistical expertise and contributions. Review statement (to acknowledge the editor and referees)

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
