# Peer review of "Storm frequency, magnitude, and cumulative storm beach impact along the U.S. east coast"

_EGUsphere, 2024_

## Author Comment (AC2)

We thank the referee for the helpful feedback.  Please see individual responses to comments below.

1- Introduction : Although the introduction is interesting and well-written, it primarily presents general information about global cyclone dynamics and lacks specific attention to the regional context and key concerns. Only a few lines in the entire introduction provide an overview of the region of interest. It would be more appropriate to concentrate on the US East Coast or at least the North Atlantic in the introduction.

The Introduction provides background information on the methods used to assess storminess organized by modeling studies followed by empirical studies. While some of these studies are global in nature, we also focus in on the relevant North Atlantic basin and U.S. east coast.

But we agree the Introduction would benefit from more localized context. To that end, we added this paragraph to the Introduction (at line 98) that transitions the Intro from general methodological background to more regional geographic context:

At the spatial scale of the U.S. east coast and centennial temporal scale, natural and potential anthropogenic forcings (e.g., sea-level rise and storms) threaten increasing populations and coastal development and ecosystems, especially given the geographic position of the U.S. coastline relative to extratropical and tropical storm tracks (e.g., Davis and Dolan, 1994; Friedman et al., 2002; Dinan 2007; Little et al., 2015; Doran et al., 2021). While much is known about the rates, spatial distribution, and acceleration of sea-level rise along the U.S. east coast during the twentieth- and twenty first-centuries (e.g., Sallenger et al., 2012; Ezer, 2013; Ezer et al., 2013; Yin and Goddard, 2013; Harvey et al., 2021; Chi et al., 2023; Yin, 2023) and changes to the wave climate over decadal time scales (e.g., Davis et al., 1993; Bromirski and Kossin, 2008; and Komar and Allan, 2007), less is known about changes to the storminess (frequency and changes in strength) over longer coastal reaches and time scales – especially using empirical data. Zhang et al. (2000) investigated water level data from 10 tide gauges from Florida to Maine and found no discernible long-term trend in the number and intensity of moderate and severe coastal storms during the twentieth century.

We also added this sentence at the end of the Introduction (line 116) to provide additional rationale for using SEPI:

A primary advantage of using this method is that sea-level change (i.e., rise) is removed to isolate the impact of storms on beach erosion potential and therefore, a rise in sea-level will exacerbate identified beach erosion potential stemming from storm tides and storm surges.

In addition, considering that numerous storm erosion predictive indices exist, it is important to clarify why SEPI and CSII were chosen, what unique contributions they offer, and what their limitations are.

We added a sentence in the Introduction (line 101) to identify the metric we use for identifying storms:

This study updates Zhang et al. (2000, 2001) Storm Erosion Potential Index (SEPI) assessment of storminess along the U.S. east coast and uses a newly developed index to assess the cumulative impact of storminess (timing and magnitude) on potential beach erosion along the U.S. east coast (Fenster and Dominguez, 2022). Like Zhang et al. (2000, 2001), we use water level data (storm tide and storm surge) to identify storms (rationale provided in 2.1 Storm Identification).

Note this sentence also refers the reader to the justification and rationale for choosing SEPI and its limitations found after line 133 in Section 2.1:

Recent studies have shown that wave runup (swash and setup processes) can contribute to extreme water levels and can induce spatially varying erosion impacts along coastlines due to varying continental shelf widths (Stockdon et al., 2007, 2023; Parker et al., 2023). However, Cohn et al. (2018) used new field datasets and a numerical model to show that anomalously high still water levels (caused by storm surge or spring tides) have a greater potential to produce dune erosion than the largest wave energy. Additionally, the effect of storm surge is purported to be larger (and the wave-driven component smaller) on the U.S. east coast than the west coast because the narrower continental shelves on the west coast limit storm surge (and enhance wave energy) more than the wider east coast shelves (Cohn et al., 2018). Serafin et al. (2017) found that slight increases in wave runup and a doubling of storm surge contribute to increases in extreme total water level events and make the case that the storm surge (high-frequency residuals) can have a 10-fold greater effect on beach

erosion on the east coast than the west coast during large storms. While SEPI and water level data do not account for potential wave runup (Stockdon et al., 2007; 2023), Zhang et al. (2001) found a linear relationship between extreme storm surges and storm waves (wave heights > 2 m) indicating that storm surges make excellent surrogates for storm waves in representing the strength of large storms. The use of storm surge data over wave data is further motivated by the reliability and long-term availability of water level, storm tide, and storm surge data.

The importance of the cumulative storm impact index (CSII) was described in Fenster and Dominguez (2022). CSII is a model that can use any storm metric. To clarify in this paper we added text to the Introduction (line 100):

This study updates Zhang et al. (2000, 2001) Storm Erosion Potential Index (SEPI) assessment of storminess along the U.S. east coast and uses a newly developed cumulative storm impact index (CSII) to account for the timing (clustering) and strength of previous storms ,to assess the cumulative impact of storminess (timing and magnitude) on potential beach erosion along the U.S. east coast (Fenster and Dominguez, 2022).

and added a sentence after CSII is introduced (line 171):

This index accounts for the timing and strength of previous storms, which make beaches more vulnerable to continued erosion (Fenster and Dominguez, 2022).

2- Method : The SEPI is calculated from $S_{2SD}$, representing the storm surge above the threshold for detecting storm surges, which is set at two standard deviations, and with a duration of 12 hours. The choice of two standard deviations and a duration of 12 hours is based on previous research. If the threshold were changed to 1.5 or 3 standard deviations or if a different duration were selected, the results would likely be affected. The choice of threshold and duration can influence the identification and quantification of storms, potentially altering the frequency and magnitude trends observed. Therefore, it is crucial to assess the robustness of the results and consider the sensitivity of the findings to different threshold and duration choices.

There was a mistake: there is no minimum duration for a storm. We have corrected the manuscript on line 392 of the discussion (additions are underlined): "We used the Storm Erosion Potential Index (SEPI) to provide thresholds for storm surges and tides that defined a storm by extreme water levels  (Zhang, 1998; Zhang et al., 2000, 2001)." Following Zhang 2000, there is a criterion of 12 hours to distinguish storms: if the interval between storms is more than 12 hours, they were taken to be distinct storms.

We did not perform a sensitivity analysis of surge threshold (or of other thresholds used to identify storms) and personnel changes have made this task unfeasible. Rather, we used established criteria to identify a storm as the *definition* of a storm, and the results stand on their own using this definition. While a sensitivity analysis of each threshold would make for a very thorough investigation, our results are based on sound rationale, are consistent with previous research (e.g., Zhang et al., 2000, 2001 found a linear relationship between 2s of the storm surge and large waves, $H_s > 2m$), and provide reasonable results (not identifying too many or too few storms relative to named storms, see Figures 10 and 11).

While the methodology for the CSII is presented in the article by Fenster and Dominguez (2022), it would be beneficial for readers if the method were further elaborated in the manuscript. For example, the justification for choosing the exponentially decaying weighting factor and the selection of tc (time constant) as one year for beach systems on the U.S. East Coast should be provided. Additionally, the determination of the delta parameter should be explained, as it plays a role in quantifying the impacts of storm clustering and large magnitude storms on sandy beaches. Justifying these choices would enhance the understanding of the methodology and the interpretation of the CSII results.

We have made the following changes in the manuscript to clarify these decisions:

Line 178: "Assuming that the recovery rate is proportional to the amount of erosion, we use an exponentially decaying weighting factor for $W_i$ (Fenster and Dominguez, 2022) where:…"

Line 188: While an appropriate value of the characteristic time, $t_c$ is crucial to understanding the meaning of the weighting function, mathematically the two parameters $t_c$ and $\delta$ may be combined into one parameter to achieve the appropriate

behavior of CSII.  See Fenster and Dominguez (2022) for additional details.  A reasonable choice of parameters will show accumulation due to storms clustered in time and will show beach recovery (CSII decreasing towards 0) when storms are temporally distant.  In practice, there are a range of parameter values that satisfy these conditions and show robust cumulative behavior, though the absolute values of CSII will fluctuate with specific parameter choices.  In this comparative study, we choose a value of $t_c$ = 1 year corresponding to the winter-summer beach profile cycle for beach systems on the U.S. east coast, and $\delta=0.3$ for consistency across all tidal gauges studied.

The estimation of $PCTE$ ($t$) is conducted over the period from 1983 to 2001. The specific choice of this time period should be justified to provide a clear rationale for the selection. Additionally, if the analysis did not include the consideration of seasonal and interannual variations of tidal components, it is crucial to explain the reason behind this decision. Providing this clarification will enhance the transparency and facilitate the interpretation of the $PCTE$ ($t$) estimates.

We chose to pull the PCTE(t) values from the NOAA database, rather than calculate them ourselves, because we readily acknowledge that NOAA's collective expertise in this area far exceeds our own.  So, we did not produce those estimates, nor did we make the decisions about how they were calculated or add any additional variations to NOAAs calculations.  We believe NOAA's data to be appropriate for our calculations.

The time period from 1983 to 2001 is the current National Tidal Data Epoch as determined by NOAA and referred to as the Current Tidal Epoch (CTE) in our manuscript.  NOAA centers ALL estimates of PCTE about the MSL of the CTE (rather than the epoch associated with the date of the data).  This was not clear to us (the authors) initially, and we appreciate the clarification that we received via personal communication with Todd Ehret of NOAA (cited in the manuscript).  NOAA keeps a set of "harmonic constituents" to reconstruct PCTE values at the time that the user requests them by plugging these constituents into a harmonic equation.  This equation, though, also requires a parameter to set the average (zero) of the water levels at some chosen value.  NOAA chooses this value to be the MSL of the CTE (1983-2001), even if we are querying dates before 1983 or after 2001.  Therefore, the PCTE levels are not comparable to the verified water levels outside of the CTE.  To calculate the correct

value of the storm surge, we had to recenter the predicted water levels (PCTE) on the MSL corresponding to the date of the data.  This is the purpose of Eq. 7.  This rationale is further explained in the paragraph that begins on Line 241.

To clarify that PCTE(t) data is pulled from NOAA, we have added on Line 213: For each station, we retrieved the following data from NOAA (McManus et. al., 2023a, 2023b; Table 1):

Although the 37 harmonic constituents (listed here: https://tidesandcurrents.noaa.gov/harcon.html?id=9410170) are ostensibly astronomical (and hydrodynamical) in nature, they do incorporate meteorological variations.  For example, the harmonic constituents Sa and Ssa7 are determined by seasonal weather changes.  Here is a relevant section from NOAA's publication "Tidal Analysis and Prediction" (https://tidesandcurrents.noaa.gov/publications/Tidal_Analysis_and_Predictions.pdf), p.119:

"… the energy at the one cycle per year (Sa) and one cycle per half year (Ssa) found by the analysis is actually meteorological in origin, namely, caused by the seasonal changes in wind, temperature, and atmospheric pressure that affect water level."

https://tidesandcurrents.noaa.gov/publications/Tidal_Analysis_and_Predictions.pdf

It short, NOAA's 37 parameter fit is already VERY good and does incorporate seasonal changes.  The method does not require additional corrections.

3- results/discussions :

Figure 5b: Do the results in terms of significance remain the same if a low-pass filter of 3-5 years is applied?

To check this, we performed an analysis of data without years excluded (that is, we use partial data rather than interpolate missing data points as required by a standard low pass filter for discrete datasets). Applying a Butterworth low pass filter of 4 years to these data showed that (1) most slopes are very close to the slopes in Table 2 and (2) the p-value for most stations improved or stayed the same.  The analysis showed all but 2 stations have statistically significant results (using $p \leq 0.05$).  (The two that did not were Newport and Montauk.  Newport's low slope was not statistically significant in

our original analysis.  The Montauk p-value went up slightly but was already just near the p-value cutoff for statistical significance.) Overall, this suggests that our results are conservative for most stations.  Results are below:

| Station | Slope (m²h/yr) (from manuscript) | p value (from manuscript) | Slope (m²h/yr) (with low pass filter) | p value (with low pass filter) |
|---|---|---|---|---|
| Portland | 0.015 | 0.082 | **0.017** | 0.003 |
| Boston | 0.017 | 0.055 | **0.012** | 0.040 |
| Newport | 0.004 | 0.363 | 0.006 | 0.853 |
| Montauk | **0.035** | 0.049 | 0.021 | 0.054 |
| The Battery | **0.045** | 0.002 | **0.044** | <0.001 |
| Sandy H. | 0.028 | 0.063 | **0.028** | 0.007 |
| Atlantic C. | **0.033** | 0.016 | **0.033** | <0.001 |
| Sewell's P. | **0.080** | 0.001 | **0.084** | <0.001 |
| Wilmington | **0.058** | 0.023 | **0.058** | <0.001 |
| Charleston | **0.023** | <0.001 | **0.023** | <0.001 |
| Fernand. B. | **0.020** | 0.019 | **0.020** | 0.002 |
| Key West | **0.028** | 0.001 | **0.027** | <0.001 |

Figure 6: What is the significance of error bars? Are the results presented over the same time period? If not, are the values comparable? It would be helpful to specify this in both the figure caption and the text.

Figure 6 is simply the average and standard deviation of all data in Figure 5.  To clarify this, we have made the following change to the caption of Figure 6 (line 318):  Average and standard deviation of each data set in Fig. 5.  Calculations include all years of data plotted in Fig. 5, and similarly  exclude years for which >=10% of data are missing.

We hope this makes it clear that the error bars are simply meant to visually identify the variation of the data in Fig. 5.  Similarly, the data included in the calculation is clear from Fig. 5.  (The overall ranges of the periods of record are also listed in Table 1, but Figure 5 shows precisely which years have been excluded due to lack of data.)  Because we are characterizing each individual station, we chose to use the entire data sets of Fig. 5, rather than restrict the data to a common range.

Line 330 : "the CSII peaks appear to have a periodicity on the order of 3-10 years" Is there any explanation for this observation?

We provided a possible explanation beginning in line 449: "These aperiodic clusters have been thought to correspond with interdecadal to decadal scale variability observed in cyclonic development caused by North Atlantic Oscillation (NAO) and El Niño Southern Oscillation (ENSO) phases (Figs8, 9, 10, and 11; Davis et al., 1993; Zhang et al, 2000; Hirsch et al., 2001; Colle et al., 2015)."  However, we do agree that additional work on this topic would make an interesting future study.

Line 448-451 : Is there any variation in the distribution of the **required** recovery time throughout the observation period? Do certain stations **require** more or less time for recovery? As the authors pointed out, the time spans associated with beach recovery range from 3 years to >10 years, depending on the variability of storms in both time and space. It would be interesting to further develop this aspect, particularly in relation to existing studies in geomorphology if available.

Yes, observation of our results suggests there is variation in time, but it's not predictable. Some places are more periodic (e.g., Sewells Point) and some places are less periodic (e.g., Wilmington). Some appear more periodic in more recent times (e.g., Charleston).

With respect to recovery time for certain stations... the answer is the same, some stations would have larger and some would have smaller recovery times.

The informal examination of our results indicate that firm answers would require additional quantitative analyses and comparison to other possible explanatory data which are beyond the scope of our study. We agree with the reviewer that it would be interesting to further develop this aspect of our work and relate it to geomorphology studies (especially NAO and ENSO events) as a standalone project.

Additionally, it seemed to us that the questions indicated a slight misunderstanding of our method. To clarify, we changed the language in the text (line 450) from "require" to "allow" indicating that a time period exists within which recovery can occur and not the actual recovery:

The results from this study show that peaks and troughs tend to vary on time scales of four to 10 years and provide insight into the time scale  allowed for beaches to "heal" after storm clusters and large magnitude storms occur (Figs. 8, 9, 10, and 11).

Line 451-454 : It would be interesting to investigate whether these aperiodic clusters truly correspond to the interdecadal to decadal scale variability observed in cyclonic development attributed to the North Atlantic Oscillation and El Niño Southern Oscillation phases.

We agree, see above.

---

## Author Comment (AC3)

We thank the reviewer for the careful reading of the manuscript and thoughtful feedback. Please see below for replies to specific comments.

1.  I don't agree with some of the jargon used in the title and in the paper's findings related to "storm" and "storm surge." The definition of "storm" comes from the thresholding of extreme water level and non-tidal residual (defined as storm surge here) used in the SEPI, and the "storm magnitude" is the magnitude of the index, NOT of the water level variables.

The referee's distinction between "storm" and "storm surge" is correct, and we agree that some of the manuscript's language is descriptive rather than technical.  We have made some clarifying changes and itemized them in responses below.

Similarly, the title suggests the paper is evaluating storm surge frequency, yet the variable that is evaluated is not storm surge. While a high storm surge is necessary for causing the SEPI, the storm tide also has to meet a threshold. This means there is the potential that not all storm surge events on record are analyzed.

For example, there could be events that don't cross the water level boundary, if the duration was low or the tide was low. So the suggestion the paper is evaluating storm surge frequency and magnitude (which is the elevation of the storm surge) is misleading.

We agree that the title is misleading and have changed it from: "Storm surge frequency, magnitude, and cumulative storm beach impact along the U.S. east coast" to "Storm frequency, magnitude, and cumulative storm beach impact along the U.S. east coast."

That said, there are inconsistencies in jargon throughout – using SEPI storm magnitude, average annual SEPI, storm magnitude, SEPI, and SEPI Values, SEPI measures of storm magnitude, all to describe the same value.

Each storm has a single value for SEPI, calculated according to Eq 1.  SEPI is defined as the storm magnitude.  So, while SEPI refers to the calculation in Eq 1, the terms "storm magnitude" (lines 272, 297, 321, 394, 405, 420, and 434) and "measures of storm magnitude" (line 354) are descriptive and used to remind the reader of the meaning of SEPI.  We believe that this adds to the clarity and (in some cases) the grammatical flow of the language.

The "average annual SEPI," however, is NOT the same as the SEPI.  While each storm has one calculated value of SEPI (Eq. 1), the average annual SEPI is calculated for each year from the SEPI values of individual storms occurring in that year.  It is the average of all SEPI storm values for a given year.

We have clarified this distinction by making the following changes (additions are underlined):

- Line 154: The SEPI storm magnitude for a single storm is defined in terms of these two thresholds:

- Line 291: Added the sentence: Figure 5b shows the SEPI averaged over all storms each year for each tidal station.
- Line 304 in Figure caption 5: Annual results of (a) storm frequency (number of storms/year) and (b) storm magnitude (SEPI averaged over all storms identified in each year) arranged from north (top) to south (bottom).
- Line 318 in Figure Caption 6:  Average and standard deviation of each data set in Fig. 5.    Calculations include all years of data plotted in Fig. 5, and similarly  exclude years for which ≥10% of data are missing.

2.   The authors justify the lack of inclusion of waves by stating storm tide and duration as primary factors contributing to beach erosion from older studies, but many studies since then (e.g., Stockdon et al., 2007; Stockdon et al., 2023, Cohn et al., 2019, to name a few) show that wave runup (swash and setup processes) are important for spatially varying erosion impacts along coastlines. Other studies have suggested that wave runup/setup can be a large contributor to extreme water levels at the coast (e.g., Parker et al., 2023; Serafin et al., 2017; Stockdon et al., 2023). A brief discussion of the importance of these processes and potential for missing impacts is important.

Zhang et al. (2001) use the results from 11 studies, in part, to justify that storm induced beach erosion is "much more strongly related to storm tide than storm wave height." Additionally, Zhang et al. (2001) made the elaborate case that any beach erosion index should include **storm tide**, **storm wave energy**, and **storm duration**. However, while long-term storm tide data are readily available for the U.S. East Coast, long-term wave records do not exist, and wave records do not coincide with hourly water level data. This lack of empirical data makes it difficult to parse the relative contributions of the various forcings that control the total water level (TWL), i.e., waves, tides, and nontidal residuals (including storm surge) and drive storm-induced beach and/or dune erosion. Therefore, Zhang et al. (2001) used empirical and modeling studies available at the time of their publication to provide rationale and justification for using storm surge to represent storm strength and to be a surrogate for storm wave energy (e.g., Edelman, 1968, Edelman, 1972; Wood, 1982; Balsillie, 1986;  Dean, 1991; Hughes and Chui, 1981; Vellinga, 1982, 1986; Steetzel 1991, 1993; Balsillie, 1999). Additionally, empirical data demonstrated a strong linear relationship exists between hindcast significant wave heights and storm surge heights (Zhang et al., 2001). Using storm surges greater than $2\sigma$ of the annual surge level and wave heights larger than 2m, Zhang et al. (2001) suggested that this linear relationship indicates that storm surges make excellent surrogates for storm waves in representing the strength of large storms.

We acknowledge that spatial variation in beach erosion exists (as described in the suggested references). In fact, the variation exists at many different spatial scales. We also acknowledge that the relative importance of wave runup/setup varies at different spatial scales. To address these concerns, we added a paragraph in the manuscript after line 130 which provides

additional rationale for using storm surge as a proxy for waves and incorporates the suggested recent references.

Recent studies have shown that wave runup (swash and setup processes) can contribute to extreme water levels and can induce spatially varying erosion impacts along coastlines due to varying continental shelf widths (Stockdon et al., 2007, 2023; Parker et al., 2023). However, Cohn et al. (2018) used new field datasets and a numerical model to show that anomalously high still water levels (caused by storm surge or spring tides) have a greater potential to produce dune erosion than the largest wave energy. Additionally, the effect of storm surge is purported to be larger (and the wave-driven component smaller) on the U.S. east coast than the west coast because the narrower continental shelves on the west coast limit storm surge (and enhance wave energy) more than the wider east coast shelves (Cohn et al., 2018).  Serafin et al. (2017) found that slight increases in wave runup and a doubling of storm surge contribute to increases in extreme total water level events and make the case that the storm surge (high-frequency residuals) can have a 10-fold greater effect on beach erosion on the east coast than the west coast during large storms. While SEPI and water level data do not account for potential wave runup (Stockdon et al., 20007; 2023), Zhang et al. (2001) found a linear relationship between extreme storm surges and storm waves (wave heights > 2 m) indicating that storm surges make excellent surrogates for storm waves in representing the strength of large storms. The use of storm surge data over wave data is further motivated by the reliability and long-term availability of water level, storm tide, and storm surge data.

3.  How is storm duration computed? It seems important to the computation of the SEPI. It seems that the SEPI may be the sum of all hourly data over the MHW threshold for the surge "event" but this isn't explicitly stated, beyond interpretation of eqn (1). Line 166 says that there is no minimum time duration for a storm, but Line 392 says a storm needed to persist for a minimum of 12 hours.

This was a mistake: there is no minimum duration for a storm.  We have corrected the manuscript on line 392 of the discussion: "We used the Storm Erosion Potential Index (SEPI) to provide thresholds for storm surges and tides that defined a storm by extreme water levels  (Zhang, 1998; Zhang et al., 2000, 2001)."  Following Zhang 2000, there is a criterion of 12 hours to *distinguish* storms: if the interval between storms is more than 12 hours, they were taken to be distinct storms.

The storm duration is the time between the first and last data point of the sum in Eq 1.  Data that exceeds both thresholds (and therefore contributes a non-zero term to the sum in Eq 1) are grouped together as a single storm (and comprise the terms of the sum in Eq 1) when they are clustered within 12 hours of each other.

To clarify this, we have added the following to line 166: "Terms in the sum of Eq. 1 will be zero unless both thresholds are met.  Data that exceed both thresholds are grouped together as a single storm (and comprise the terms of the sum in Eq 1) when they are clustered within 12 hours of each other.  In other words, distinct storms must be separated by 12 or more hours.

The duration of the storm is the time difference between the first and last terms of the sum in Eq. 1, and there is no minimum  duration required for a storm.

4.  How is the scaling factor, f chosen for weighting beach recovery, and how much does this choice impact the model result? How sensitive is the periodicity of beach recovery to cumulative storm impacts to the parameters chosen? Is 1 year a good approximation for beach systems along gradient that may experience both ETC and TCs?

Choosing a value for f (which is equivalent to choosing a value for delta) can be done for a single tidal station to characterize the rate at which recovery or return to equilibrium occurs.  A good choice of f will show reasonable accumulation (as opposed to "artificial accumulation" described in appendix B of Fenster and Dominguez, 2022) due to storms clustered in time and will show beach recovery (CSII decreases towards 0) when storms are temporally distant.  In practice, there are a range of f values that satisfy these conditions, and over this range of reasonable f values, the periodicity of beach recovery (observed in say, Fig. 8) does NOT change. (What does change is the overall range of CSII values: the peaks of CSII values may get higher or lower, but their positions in time do not change.  Therefore, periodicity also does not change.) Thus, the overall results presented are robust relative to the choice of f value.  More detail on the choice of f (or delta) value, along with sample data for f values chosen too high and too low are given in our previous paper (Fenster and Dominguez, 2022), especially Appendix B.

There is also more detail on the interpretation of t_c in our previous paper (Fenster and Dominguez, 2022), especially Section 3.   It makes sense to choose an appropriate t_c to more easily interpret the weighting function; i.e., at time t_c after a storm (that is, t_p=t_c and tau_p=1) the beach has *mostly* recovered to its equilibrium state. Additionally, one could really *calibrate* the value of t_c for an individual tide gauge by validating data for a nearby beach.

But in practice, the index is robust enough to correct for an "incorrect" choice of t_c through the appropriate choice of f. (Note that mathematically, the parameters f and t_c can be redefined as a single parameter.)  And in this study, which focuses on comparing 12 different tidal gauges, we did not calibrate any single location, but rather chose one set of parameters (t_c = 1 and delta = 0.3) that were reasonable for the entire set of gauges.

To clarify, we have added the following at line 188: While an appropriate value of the characteristic time, $t_c$, is crucial to understanding the meaning of the weighting function, mathematically the two parameters $t_c$ and $\delta$ may be combined into one parameter to achieve the appropriate behavior of CSII (See Fenster and Dominguez (2022) for additional details.)   A reasonable choice of parameters will show accumulation due to storms clustered in time and will show beach recovery (CSII decreasing towards 0) when storms are temporally distant.  In practice, there are a range of parameter values that satisfy these conditions and show robust cumulative behavior, though the absolute values of CSII will fluctuate with specific parameter choices.  In this comparative study, we choose a value of $t_c$ = 1 year corresponding to the winter-summer beach profile cycle for beach systems on the U.S. east coast, and $\delta=0.3$ for consistency

across all tidal gauges studied.

**Line by line**

Line 45: Typo after intensities ")"

Corrected

Line 97: Seems like Stockdon et al., 2007 would be a good reference to include here too which built off the Sallenger, 2000 publication.

We agree and have added the suggested reference. Note this reference was also used earlier in the paragraph.

Line 103: Nuance here, but I disagree the authors are assessing the frequency of and magnitude of TC and ETCs, as they're evaluating water levels, which aren't necessarily descriptive of JUST the storm climatology.

We agree this sentence is misleading and have clarified it. We are not claiming that we are analyzing TCs and ETCs. Rather we are making an interpretation based on correlation, not causality. We're using our definition of a storm and showing what this definition identified as storms and comparing those results to what's known about and consistent with storm climatology.

To clarify, we made this change in line 101: In particular, we assess the frequency and magnitude of storms along the eastern U.S. coast using historical data from 12 tidal gauge stations located from Portland, Maine to Key West, Florida and compare these results to known storm climatology of tropical and extratropical cyclones.

Line 135: Shouldn't it just be SEPI, rather than "SEPI storm index"? Otherwise, you're really saying Storm Erosion Potential Index storm index.

We agree and have deleted "storm index."

Lines 208 – 210: The Wilmington and Battery stations might also be subject to river discharge within the non-tidal residual/storm surge signals.

We agree and have added the following language at line 208 to clarify:

It should be noted that these two stations are most likely subject to tidal wave transformation caused by interactions with complex channel geometries of shallow estuaries and/or fluvial processes not prone to occur at stations located along the open ocean coast (e.g., Aubrey and Speer, 1985; Speer and Aubrey, 1985; van Rijn, 2011; Hein et al., 2021).

Line 226 – 227: While the justification that the selection of storms with MHW vs MHHW is similar is positive, a quantification of how the SEPI or duration of events is affected could be important. I believe MHHW was justified as a threshold in the original paper to infer more wave attack on dunes/back barrier from the storm, and does this relationship hold for MHW?

Yes, Zhang's original justification for using MHHW as a threshold is that waves exceeding the MHHW level will be at an elevation high enough to directly attack dunes. We do not expect this relationship to hold *in general*, especially for mixed semi-diurnal systems such as tidal systems along the US West Coast. We rely only on the similarity of the MHW levels to the MHHW levels for the tidal gauges investigated along the semi-diurnal US east coast as well as the sensitivity analysis mentioned in our paper. To make this more clear, we have made the following additions to the paragraph beginning on line 221:

We note that Zhang et al. (2000, 2001) relied on the condition that water levels exceed the mean higher high water (MHHW) value, rather than the MHW, to identify storms. This is because storm tide above MHHW is high enough to directly and forcefully attack the dunes (Zhang, 2001). Because the U.S. east coast experiences largely semi-diurnal tides and because the difference between MHW and MHHW is small, it was not standard practice for NOAA's National Ocean Service to calculate historical MHHW values at tide gauges located on the U.S. east coast (T. Ehret, NOAA, personal communication). Furthermore, we conducted sensitivity analyses to determine the differences between MHW and MHHW for more recent years when water level data from both datums were available. These analyses revealed no significant differences in storm identification results for the stations considered using MHW compared to MHHW. It should be noted, however, that MHW should not replace the MHHW threshold in general. It is not expected that the MHW level is high enough for waves to do significant work on dunes for mixed semi-diurnal systems such as the the U.S. West Coast.

Line 262: How was the standard deviation over time computed? (if saying they are constant must have looked at time variability in this parameter?)

The data used for this analysis was all *hourly* storm surges available over the entire period of record for each station. We calculated the average and standard deviation of these data for each station over the entire period of record and plotted each distribution to verify that each was Gaussian distributed.

To clarify this in the manuscript, we have changed line 260: "For all 12 stations, the hourly average storm surge datas are approximately Gaussian distributed, centered about a value of zero..."

To look for changes in time, we divided the data (over the entire period of record) into 9 time periods for each station. For Wilmington, NC, the period of record is from December 1935 -

December 2022 (87 years). Therefore, each smaller time period consisted of data over about a 10-year period. From these data sets, we calculated the mean and standard deviation of the hourly surges. We found that the standard deviations were approximately constant over the time periods investigated. The only large deviation in surge distributions over time that we identified was for Wilmington, NC (as identified in the manuscript.) The standard deviations quoted in the paper come from this statistical analysis over the 10-year time periods.

We did not think that most of these details were helpful for the paper, but we have changed lines 261-262 to include more detail for clarity: "The standard deviation of the hourly surge values, $\sigma_s$,  measured over approximately 10 year time intervals, do not change appreciably over time for almost all stations with values ranging from $\sigma_s$ = 0.07 m (Key West) to $\sigma_s$ = 0.17 m (The Battery and Sandy Hook)."

MHW threshold takes into account sea level rise, why not just remove the MSL trend from the data and use a stationary MHW threshold? How does sea level rise effect results? Or is the inclusion of a time varying MHW/MSL take care of that?

In short, we use a single *surge* threshold because surge distributions do not vary over the time period of record (except Wilmington to some extent, as noted in the manuscript), but we use a time varying *MHW* threshold because the MHW *does* vary in time. Sea-level rise accounts for the long-term upward trend in MHW values, but there are also annual variations. The main reason that we use a moving threshold instead of a single threshold is so that our threshold is more accurate over shorter timescales. Annual calculations of the MHW threshold will more accurately reflect the MHW level for the beach at any particular time compared to a single threshold, and therefore should better predict erosion potential. (That said, we DID test a single MHW threshold and did not find significant differences in the results.)

Yes, the inclusion of a time varying MHW threshold removes the effect of sea-level rise. If we do NOT account for sea level rise (use a single MHW threshold and do NOT adjust the verified water levels), we would get SEPI values that increase dramatically over the period of record. While this DOES reflect increased damage to beaches which are being hit at much higher levels due to sea level rise, it will overestimate the frequency and magnitude of storms during later years and underestimate the frequency and magnitude during earlier years.

The suggestion to remove the MSL trend from the data is certainly an option. We could have subtracted the annual MSL from all verified water level data and used a single threshold. To determine that single threshold, we would also have had to subtract the MSL from the MHW data and find the average of the MHW over the entire period of record. However, our approach also removes MSL rise, uses an appropriately varying threshold (as described above), has the advantage of staying close to the raw data, and follows established protocol of Zhang et. Al, 2000 and 2001.

Line 365 – 370: Hurricane Florence impacted the Carolinas with an incredible amount of precipitation too. Is the signal Wilmington seeing due to river flow rather than coastal driven storm surge? Especially if potentially the duration of the event was impacted, e.g., gauge water levels staying high for much longer due to river flow outletting post storm surge event.

We agree and added language at line 369 to clarify:

The tide stations to the south of Wilmington did not record these storms appreciably. Despite the particularly stormy July 2018 along the North Carolina coast, we note that the peak stream stage measured at the stream gauge in closest proximity to (and upstream from) the Wilmington tide gauge (Cape Fear R at Lock 1 NR, Kelly, NC, 02105769) was approximately 11.6% greater than the mean stage (5.4 m vs. 4.9 m, respectively) and 15.5% greater than the median stage (5.4 m vs. 4.7 m, respectively). Consequently, post-storm river flow most certainly impacted the Wilmington water level data by approximately 11-16%.

Line 390: Again, not looking at spatial and temporal trends in storm surge/storm tide, looking at trends in SEPI

We have made the following changes at line 389 for clarity: "This study extracted historical water level data from 12 NOAA tide gauge stations, spanning the early 20th century to 2022 from central Maine to southern Florida, to determine if temporal and spatial trends existed in  storm  frequency and magnitude along the U.S. Atlantic Ocean coast."

Line 396: What are the typical problems associated with empirical data analyses the authors are referring to?

The typical problems are discussed in the intro lines 68-70, but we have reiterated at at line 396 as well:

Our methods avoid typical problems associated with empirical data analyses such as using heterogeneous historical instrumental data and limited temporal data to detect long-term (decadal to centennial) trends in cyclone frequency or intensity.

In Figure 1, what is considered the duration? This might be a good place to include it

We agree that it would be helpful to illustrate the duration of the storm in a figure.  However, Figure 1 only illustrates the first threshold, while Figure 2 illustrates the second.  In order to qualify as data that contribute to a storm, BOTH thresholds must be met.  Therefore, we cannot point to the beginning and end of the storm on either figure (since it is the product of these two data that identify the storm) and must rely on the mathematical description.  As noted above, we have added language at line 166 to clarify what we mean by the duration of the storm.

---

## Author Response (AR2)

We thank the referee for the helpful feedback. Please see individual responses to comments below.

Additional comments made in response to the associate editor's email dated 8/8/24 and changes made to the manuscript are highlighted in blue text and yellow highlight.

1- Introduction : Although the introduction is interesting and well-written, it primarily presents general information about global cyclone dynamics and lacks specific attention to the regional context and key concerns. Only a few lines in the entire introduction provide an overview of the region of interest. It would be more appropriate to concentrate on the US East Coast or at least the North Atlantic in the introduction.

The Introduction provides background information on the methods used to assess storminess organized by modeling studies followed by empirical studies. While some of these studies are global in nature, we also focus in on the relevant North Atlantic basin and U.S. east coast.

But we agree the Introduction would benefit from more localized context. To that end, we added this paragraph to the Introduction (at line 98) that transitions the Intro from general methodological background to more regional geographic context:

At the spatial scale of the U.S. east coast and centennial temporal scale, natural and potential anthropogenic forcings (e.g., sea-level rise and storms) threaten increasing populations and coastal development and ecosystems, especially given the geographic position of the U.S. coastline relative to extratropical and tropical storm tracks (e.g., Davis and Dolan, 1994; Friedman et al., 2002; Dinan 2007; Little et al., 2015; Doran et al., 2021). While much is known about the rates, spatial distribution, and acceleration of sea-level rise along the U.S. east coast during the twentieth- and twenty first-centuries (e.g., Sallenger et al., 2012; Ezer, 2013; Ezer et al., 2013; Yin and Goddard, 2013; Harvey et al., 2021; Chi et al., 2023; Yin, 2023) and changes to the wave climate over decadal time scales (e.g., Davis et al., 1993; Bromirski and Kossin, 2008; and Komar and Allan, 2007), less is known about changes to the storminess (frequency and changes in strength) over longer coastal reaches and time scales – especially using empirical data. Zhang et al. (2000) investigated water level data from 10 tide gauges from Florida to Maine and found no discernible long-term trend in the number and intensity of moderate and severe coastal storms during the twentieth century.

We also added this sentence at the end of the Introduction (line 116) to provide additional rationale for using SEPI:

A primary advantage of using this method is that sea-level change (i.e., rise) is removed to isolate the impact of storms on beach erosion potential and therefore, a rise in sea-level will exacerbate identified beach erosion potential stemming from storm tides and storm surges.

In addition, considering that numerous storm erosion predictive indices exist, it is important to clarify why SEPI and CSII were chosen, what unique contributions they offer, and what their limitations are.

We added a sentence in the Introduction (line 101) to identify the metric we use for identifying storms:

This study updates Zhang et al. (2000, 2001) Storm Erosion Potential Index (SEPI) assessment of storminess along the U.S. east coast and uses a newly developed index to assess the cumulative impact of storminess (timing and magnitude) on potential beach erosion along the U.S. east coast (Fenster and Dominguez, 2022). Like Zhang et al. (2000, 2001), we use water level data (storm tide and storm surge) to identify storms (rationale provided in 2.1 Storm Identification).

Note this sentence also refers the reader to the justification and rationale for choosing SEPI and its limitations found after line 133 in Section 2.1:

Recent studies have shown that wave runup (swash and setup processes) can contribute to extreme water levels and can induce spatially varying erosion impacts along coastlines due to varying continental shelf widths (Stockdon et al., 2007, 2023; Parker et al., 2023). However, Cohn et al. (2018) used new field datasets and a numerical model to show that anomalously high still water levels (caused by storm surge or spring tides) have a greater potential to produce dune erosion than the largest wave energy. Additionally, the effect of storm surge is purported to be larger (and the wave-driven component smaller) on the U.S. east coast than the west coast because the narrower continental shelves on the west coast limit storm surge (and enhance wave energy) more than the wider east coast shelves (Cohn et al., 2018).  Serafin et al. (2017) found that slight increases in wave runup and a doubling of storm surge contribute to increases in extreme total water level events and make the case that the

storm surge (high-frequency residuals) can have a 10-fold greater effect on beach erosion on the east coast than the west coast during large storms. While SEPI and water level data do not account for potential wave runup (Stockdon et al., 2007; 2023), Zhang et al. (2001) found a linear relationship between extreme storm surges and storm waves (wave heights > 2 m) indicating that storm surges make excellent surrogates for storm waves in representing the strength of large storms. The use of storm surge data over wave data is further motivated by the reliability and long-term availability of water level, storm tide, and storm surge data.

The importance of the cumulative storm impact index (CSII) was described in Fenster and Dominguez (2022). CSII is a model that can use any storm metric. To clarify in this paper we added text to the Introduction (line 100):

This study updates Zhang et al. (2000, 2001) Storm Erosion Potential Index (SEPI) assessment of storminess along the U.S. east coast and uses a newly developed cumulative storm impact index (CSII) to account for the timing (clustering) and strength of previous storms ,to assess the cumulative impact of storminess (timing and magnitude) on potential beach erosion along the U.S. east coast (Fenster and Dominguez, 2022).

and added a sentence after CSII is introduced (line 171):

This index accounts for the timing and strength of previous storms, which make beaches more vulnerable to continued erosion (Fenster and Dominguez, 2022).

We fully addressed all concerns of Reviewer 2, comment 1 in our previous response by adding a paragraph to the Introduction and adding clarifying language in multiple areas as shown above and in the manuscript. No additional changes required.

2- Method : The SEPI is calculated from $S_{2SD}$, representing the storm surge above the threshold for detecting storm surges, which is set at two standard deviations, and with a duration of 12 hours. The choice of two standard deviations and a duration of 12 hours is based on previous research. If the threshold were changed to 1.5 or 3 standard deviations or if a different duration were selected, the results would likely be affected. The choice of threshold and duration can influence the identification and

quantification of storms, potentially altering the frequency and magnitude trends observed. Therefore, it is crucial to **assess** the robustness of the results and consider the sensitivity of the findings to different threshold and duration choices.

There was a mistake: there is no minimum duration for a storm. We have corrected the manuscript on line 392 of the discussion (additions are underlined): "We used the Storm Erosion Potential Index (SEPI) to provide thresholds for storm surges and tides that defined a storm by extreme water levels  (Zhang, 1998; Zhang et al., 2000, 2001)." Following Zhang 2000, there is a criterion of 12 hours to distinguish storms: if the interval between storms is more than 12 hours, they were taken to be distinct storms.

We did not perform a sensitivity analysis of surge threshold (or of other thresholds used to identify storms) and personnel changes have made this task unfeasible. Rather, we used established criteria to identify a storm as the *definition* of a storm, and the results stand on their own using this definition. While a sensitivity analysis of each threshold would make for a very thorough investigation, our results are based on sound rationale, are consistent with previous research (e.g., Zhang et al., 2000, 2001 found a linear relationship between 2s of the storm surge and large waves, $H_s > 2m$), and provide reasonable results (not identifying too many or too few storms relative to named storms, see Figures 10 and 11).

To incorporate our response to this reviewer comment into the manuscript, we have added the following language at line 139:

The threshold of storm surges in excess of two standard deviations is consistent with previous research (e.g., Zhang et al., 2000, 2001) that found a linear relationship exists between two standard deviations of the storm surge and large waves ($H_s > 2m$). Additionally, using two standard deviations provides reasonable results by not identifying too many storms or too few storms relative to named storms (see Results and Supplements 1 and 2).

While the methodology for the CSII is presented in the article by Fenster and Dominguez (2022), it would be beneficial for readers if the method were further elaborated in the manuscript. For example, the justification for choosing the

exponentially decaying weighting factor and the selection of tc (time constant) as one year for beach systems on the U.S. East Coast should be provided. Additionally, the determination of the delta parameter should be explained, as it plays a role in quantifying the impacts of storm clustering and large magnitude storms on sandy beaches. Justifying these choices would enhance the understanding of the methodology and the interpretation of the CSII results.

We have made the following changes in the manuscript to clarify these decisions:

Line 178: "Assuming that the recovery rate is proportional to the amount of erosion, we use an exponentially decaying weighting factor for $W_i$ (Fenster and Dominguez, 2022) where:..."

Line 188: While an appropriate value of the characteristic time, $t_c$, is crucial to understanding the meaning of the weighting function, mathematically the two parameters $t_c$ and $\delta$ may be combined into one parameter to achieve the appropriate behavior of CSII.  See Fenster and Dominguez (2022) for additional details.  A reasonable choice of parameters will show accumulation due to storms clustered in time and will show beach recovery (CSII decreasing towards 0) when storms are temporally distant.  In practice, there are a range of parameter values that satisfy these conditions and show robust cumulative behavior, though the absolute values of CSII will fluctuate with specific parameter choices.  In this comparative study, we choose a value of $t_c$ = 1 year corresponding to the winter-summer beach profile cycle for beach systems on the U.S. east coast, and $\delta=0.3$ for consistency across all tidal gauges studied.

The estimation of $PCTE$ ($t$) is conducted over the period from 1983 to 2001. The specific choice of this time period should be justified to provide a clear rationale for the selection.

There seem to be two misunderstandings here.

First, we chose to pull the PCTE(t) values from the NOAA database, rather than calculate them ourselves, because we readily acknowledge that NOAA's collective expertise in this area far exceeds our own.  So, we did not produce those estimates,

nor did we make the decisions about how they were calculated or add any additional variations to NOAAs calculations.

To clarify that PCTE(t) data is pulled from NOAA, we have added on Line 213: For each station, we retrieved the following data from NOAA (McManus et. al., 2023a, 2023b; Table 1):

Second, the estimation of PCTE is not conducted only from 1983 to 2001.  Rather it is conducted over the entire period of record, using data from the entire period of record.  The zero value of all estimations are then SET to be the mean sea level of the CTE, which is the time period from 1983 to 2001.  NOAA centers ALL estimates of PCTE about the MSL of the CTE (rather than the epoch associated with the date of the data). This was not clear to us (the authors) initially, and we appreciate the clarification that we received via personal communication with Todd Ehret of NOAA.  This is already cited in the manuscript on line 243:

NOAA uses the most current set of harmonic constants to generate these predicted water levels $PCTE(t)$ over the entire period of the data retrieval request and sets the mean of all predicted levels to be the MSL for the current 1983-2001 tidal epoch (CTE) (T. Ehret, NOAA, personal communication).  (emphasis added.)

NOAA keeps a set of "harmonic constituents" to reconstruct PCTE values at the time that the user requests them by plugging these constituents into a harmonic equation. This equation, though, also requires a parameter to set the average (zero) of the water levels at some chosen value.  NOAA chooses this value to be the MSL of the CTE (1983-2001), even if we are querying dates before 1983 or after 2001.  Therefore, the PCTE levels are not comparable to the verified water levels outside of the CTE.  To calculate the correct value of the storm surge, we had to recenter the predicted water levels (PCTE) on the MSL corresponding to the date of the data.  This is the purpose of Eq. 7. This rationale is further explained beginning on Line 245:

Tides are expected to oscillate about the mean sea level, but NOAA's method of prediction does not account for the significant sea-level rise that occurs over the time scale of interest.  Consequently, the mean of the $PCTE(t)$ values does not match the actual mean sea level for (at least) the time periods prior to the current tidal epoch.  We correct for this by setting the mean of the predicted water levels to be equal to the actual mean sea level on any given year.

Additionally, if the analysis did not include the consideration of seasonal and interannual variations of tidal components, it is crucial to explain the reason behind this decision. Providing this clarification will enhance the transparency and facilitate the interpretation of the $PCTE$ ($t$) estimates.

Although the 37 harmonic constituents (listed here: https://tidesandcurrents.noaa.gov/harcon.html?id=9410170) are ostensibly astronomical (and hydrodynamical) in nature, they do incorporate meteorological variations.  For example, the harmonic constituents Sa and Ssa7 are determined by seasonal weather changes.  Here is a relevant section from NOAA's publication "Tidal Analysis and Prediction" (https://tidesandcurrents.noaa.gov/publications/Tidal_Analysis_and_Predictions.pdf), p.119:

"… the energy at the one cycle per year (Sa) and one cycle per half year (Ssa) found by the analysis is actually meteorological in origin, namely, caused by the seasonal changes in wind, temperature, and atmospheric pressure that affect water level."

https://tidesandcurrents.noaa.gov/publications/Tidal_Analysis_and_Predictions.pdf

It short, NOAA's 37 parameter fit is already VERY good and does incorporate seasonal changes.  The method does not require additional corrections.

To show that the analysis does include seasonal variations, we have added the following at line 242, as well as the reference cited above.

To produce the $P_{CTE}$(t) values, NOAA uses a Least Squares Harmonic Analysis at each station to produce a set of *harmonic constants,* which empirically weight 37 different harmonic constituents to account for various astronomical, hydrodynamical, and seasonal influences on the tides for each station (NOAA, 2023; Parker, 2007).

3- results/discussions :

Figure 5b: Do the results in terms of significance remain the same if a low-pass filter of 3-5 years is applied?

To address this, we performed additional analyses and added it to the manuscript as Supplement 3:

Supplement 3

To test the significance of the linear regression results presented in Figure 5b and Table 2, we performed additional analysis using a Butterworth low pass filter of four years. The data set for this analysis included years with partial data (with 10% or more missing data), rather than treat those data points as missing and interpolate them as required by a standard low pass filter for discrete datasets. The results, shown in Table S3, alongside the results from Table 2 for comparison, indicate that (1) most slope values are very close to the slope values in Table 2 and (2) the p-value for most stations improved or stayed the same. The analysis showed all but two stations have statistically significant results (using $p \leq 0.05$). (The two that did not were Newport and Montauk. Newport's low slope was not statistically significant in our original analysis. The Montauk p-value went up slightly but was already near the p-value cutoff for statistical significance.) Overall, this suggests that results in Table 2 are conservative for most stations.

Table S3: Comparison of slope and p values for annual SEPI per storm data, including results from Table 2, and results from analysis with a low pass filter of four years applied. Fit parameters for average annual SEPI per storm, corresponding to Fig. 5b. Slopes in boldface are statistically significant at $p \leq 0.05$.

| Station | Slope ($m^2h/yr$) (from Table 2) | p value (from Table 2) | Slope ($m^2h/yr$) (with low pass filter) | p value (with low pass filter) |
|---|---|---|---|---|
| Portland | 0.015 | 0.082 | **0.017** | 0.003 |
| Boston | 0.017 | 0.055 | **0.012** | 0.040 |
| Newport | 0.004 | 0.363 | 0.006 | 0.853 |
| Montauk | **0.035** | 0.049 | 0.021 | 0.054 |
| The Battery | **0.045** | 0.002 | **0.044** | <0.001 |
| Sandy H. | 0.028 | 0.063 | **0.028** | 0.007 |
| Atlantic C. | **0.033** | 0.016 | **0.033** | <0.001 |
| Sewell's P. | **0.080** | 0.001 | **0.084** | <0.001 |
| Wilmington | **0.058** | 0.023 | **0.058** | <0.001 |

| | | | | |
|---|---|---|---|---|
| Charleston | **0.023** | <0.001 | **0.023** | <0.001 |
| Fernand. B. | **0.020** | 0.019 | **0.020** | 0.002 |
| Key West | **0.028** | 0.001 | **0.027** | <0.001 |

We have also added the following at line 297:

Similar results are found when a low pass filter of four years is applied to the data in Figure 5b (Supplement 3).

Figure 6: What is the significance of error bars? Are the results presented over the same time period? If not, are the values comparable? It would be helpful to specify this in both the figure caption and the text.

Figure 6 is simply the average and standard deviation of all data in Figure 5. To clarify this, we have made the following change to the caption of Figure 6 (line 318): Average and standard deviation of each data set in Fig. 5.  Calculations include all years of data plotted in Fig. 5, and similarly  exclude years for which >=10% of data are missing.

We hope this makes it clear that the error bars are simply meant to visually identify the variation of the data in Fig. 5. Similarly, the data included in the calculation is clear from Fig. 5. (The overall ranges of the periods of record are also listed in Table 1, but Figure 5 shows precisely which years have been excluded due to lack of data.) Because we are characterizing each individual station, we chose to use the entire data sets of Fig. 5, rather than restrict the data to a common range.

Line 330 : "the CSII peaks appear to have a periodicity on the order of 3-10 years" Is there any explanation for this observation?

We provided a possible explanation beginning in line 449: "These aperiodic clusters have been thought to correspond with interdecadal to decadal scale variability

observed in cyclonic development caused by North Atlantic Oscillation (NAO) and El Niño Southern Oscillation (ENSO) phases (Figs8, 9, 10, and 11; Davis et al., 1993; Zhang et al, 2000; Hirsch et al., 2001; Colle et al., 2015)."  However, we do agree that additional work on this topic would make an interesting future study.

Line 448-451 : Is there any variation in the distribution of the **required** recovery time throughout the observation period? Do certain stations **require** more or less time for recovery? As the authors pointed out, the time spans associated with beach recovery range from 3 years to >10 years, depending on the variability of storms in both time and space. It would be interesting to further develop this aspect, particularly in relation to existing studies in geomorphology if available.

Yes, observation of our results suggests there is variation in time, but it's not predictable. Some places are more periodic (e.g., Sewells Point) and some places are less periodic (e.g., Wilmington). Some appear more periodic in more recent times (e.g., Charleston).

With respect to recovery time for certain stations… the answer is the same, some stations would have larger and some would have smaller recovery times.

The informal examination of our results indicate that firm answers would require additional quantitative analyses and comparison to other possible explanatory data which are beyond the scope of our study. We agree with the reviewer that it would be interesting to further develop this aspect of our work and relate it to geomorphology studies (especially NAO and ENSO events) as a standalone project.

Additionally, it seemed to us that the questions indicated a slight misunderstanding of our method. To clarify, we changed the language in the text (line 450) from "require" to "allow" indicating that a time period exists within which recovery can occur and not the actual recovery:

The results from this study show that peaks and troughs tend to vary on time scales of four to 10 years and provide insight into the time scale  allowed for beaches to "heal" after storm clusters and large magnitude storms occur (Figs. 8, 9, 10, and 11).

Line 451-454 : It would be interesting to investigate whether these aperiodic clusters truly correspond to the interdecadal to decadal scale variability observed in cyclonic

development attributed to the North Atlantic Oscillation and El Niño Southern Oscillation phases.

We agree, see above.

The following text was added to address the periodicity comments and concerns by this reviewer beginning at line 451:

This four-to-10-year time period varies temporally and spatially with some locations apparently more periodic in nature (e.g., Sewells Point), some places less periodic (e.g., Wilmington), and other locations more periodic in recent times (e.g., Charleston). While causative explanations for these variations are beyond the scope of this study,  the aperiodic clusters have been thought to correspond with interdecadal to decadal scale variability observed in cyclonic development caused by North Atlantic Oscillation (NAO) and El Niño Southern Oscillation (ENSO) phases (Figs8, 9, 10, and 11; Davis et al., 1993; Zhang et al, 2000; Hirsch et al., 2001; Colle et al., 2015).

… and, referring to future studies, we added to the last line of the conclusions after line 492:

Such studies should include identifying regional and local factors that control the beach recovery time and compared that to the time allowed for beach recovery based on the CSII analysis and the observed four to 10 year interdecadal variation in the observed CSII values.